# Learning from Sparse Offline Datasets via Conservative Density Estimation

**Zhepeng Cen**[1], **Zuxin Liu**[1], **Zitong Wang**[2], **Yihang Yao**[1], **Henry Lam**[2], **Ding Zhao**[1]
[1]Carnegie Mellon University, [2] Columbia University
`{zcen, zuxinl, yihangya}@andrew.cmu.edu`
`{zw2690, henry.lam}@columbia.edu, dingzhao@cmu.edu`

## Abstract

Offline reinforcement learning (RL) offers a promising direction for learning policies from pre-collected datasets without requiring further interactions with the environment. However, existing methods struggle to handle out-of-distribution (OOD) extrapolation errors, especially in sparse reward or scarce data settings. In this paper, we propose a novel training algorithm called Conservative Density Estimation (CDE), which addresses this challenge by explicitly imposing constraints on the state-action occupancy stationary distribution. CDE overcomes the limitations of existing approaches, such as the stationary distribution correction method, by addressing the support mismatch issue in marginal importance sampling. Our method achieves state-of-the-art performance on the D4RL benchmark. Notably, CDE consistently outperforms baselines in challenging tasks with sparse rewards or insufficient data, demonstrating the advantages of our approach in addressing the extrapolation error problem in offline RL. Code is available at `https://github.com/czp16/cde-offline-rl`.

## 1 Introduction

Reinforcement Learning (RL) has witnessed remarkable advancements in recent years (Akkaya et al., 2019; Kiran et al., 2021). Nevertheless, the success of RL relies on continuous online interactions, resulting in high sample complexity and potentially restricting its practical applications in real-world scenarios (Levine et al., 2016; Gu et al., 2022). As a compelling solution, offline RL has been brought to the fore, with the objective of learning effective policies from pre-existing datasets, thereby eliminating the necessity for further environment interactions (Fu et al., 2020; Prudencio et al., 2023).

Despite its benefits, offline RL is not devoid of challenges, most notably the out-of-distribution (OOD) extrapolation errors, which emerge when the agent encounters state-actions that were absent in the dataset. These issues pose significant hurdles when learning policies from datasets with sparse rewards or low coverage of state-action spaces (Levine et al., 2020). To address OOD estimation errors in value-based offline RL, current efforts primarily revolve around two strategies: pessimism-based methods (Xie et al., 2021a; Shi et al., 2022) and the integration of regularizations (Kostrikov et al., 2021a). However, these approaches hinge on assumptions of the behavior policy. In addition, many works push policy to behavior policy to achieve pessimism (Fujimoto et al., 2019; Shi et al., 2022), which is more challenging to select the level of pessimism when the data distribution estimation is difficult (Liu et al., 2020; Xie et al., 2021a) (e.g., in high-dimensional state-action space), while regularization methods may struggle with the tuning of the regularization coefficient (Lee et al., 2021; Lyu et al., 2022). As such, striking the optimal balance of conservativeness remains a challenging goal particularly in sparse-reward settings.

Recent attention has been drawn towards an alternative method that employs importance sampling (IS) for offline data distribution correction (Precup, 2000; Jiang & Li, 2016). Among these, Distribution Correction-Estimation (DICE)-based methods have garnered substantial interest, which uses a single marginal ratio to reweight rewards for each state-action pair and has a relatively low estimation variance (Nachum et al., 2019b; Zhang et al., 2020; Lee et al., 2021). DICE provides a behavior-agnostic estimation of stationary distributions, presenting a more direct approach for offline learning. However, DICE-based techniques rely on an implicit assumption of the dataset's

concentrability(Munos, 2007; Xie et al., 2021b; Li et al., 2022), otherwise the stationary distribution support mismatch between the dataset and policy can cause an arbitrarily large IS ratio, resulting in unstable training and poor performance, which can be significantly severe with insufficient data.

To address these challenges, we introduce a novel method, the Conservative Density Estimation (CDE), that integrates the strengths of both pessimism-based and DICE-based approaches. CDE employs the principles of conservative Q-learning (Kumar et al., 2020) in a unique way, incorporating pessimism within the stationary distribution space to achieve a theoretically-backed conservative occupation distribution. On the one hand, CDE does not rely on Bellman update-style value estimation, favoring a direct behavior-policy-agnostic stationary distribution correction that improves performance in *sparse reward scenarios*. On the other hand, by constraining the density of the stationary distribution induced by OOD state-action pairs, CDE significantly enhances performance in *data-limited settings*. This stands in contrast to the significant performance degradation observed in baseline offline RL methods with diminishing dataset sizes, as CDE maintains high rewards even with only **1% trajectories** in challenging D4RL tasks (Fu et al., 2020).

1. We introduce the first approach to explicitly apply pessimism in the stationary distribution space. Notably, CDE outperforms conservative value learning-based approaches in sparse reward settings and demonstrates superior performance over DICE-based methods in handling scarce data situations.

2. We present a method that automatically bounds the concentrability coefficient without resorting to the common concentrability assumption (Rashidinejad et al., 2021; Shi et al., 2022; Zhan et al., 2022), underlining its robustness in managing the OOD extrapolation issue inherent in offline RL.

3. We demonstrate the resilience of CDE in maintaining high rewards even with significantly reduced dataset sizes, such as $1\%$ of trajectories, while prior methods fail. Therefore, our method provides a viable solution for real-world applications where data can be scarce or costly to obtain.

## 2 RELATED WORK

**Offline RL with regularization or constraints.** To mitigate OOD issues, Q-value-based methods are often enhanced with regularization or constraint terms (Levine et al., 2020; Prudencio et al., 2022). These techniques restrict the learned policy's deviation from the behavior policy in the dataset, whether through explicit constrained policy spaces (Fujimoto et al., 2019) or regularizers in the objective (Wu et al., 2019; Peng et al., 2019; Kumar et al., 2019; Nair et al., 2020; Fujimoto & Gu, 2021). Alternatively, value regularization is employed to yield lower estimates for unseen states or actions, resulting in conservative policies (Kostrikov et al., 2021a; Kumar et al., 2020). However, those methods may suffer instability from approximation error when learning value with Bellman update iteratively (Fujimoto et al., 2018; Fu et al., 2019; Brandfonbrener et al., 2021), often failing in sparse reward settings even with expert demonstrations. Meanwhile, the reliance on heuristic regularization can lead to overly conservative policies and degrade performance.

**Offline RL with marginal importance sampling.** The DICE method represents a class of approaches that directly address distribution shift using marginal importance sampling, offering reduced estimation variance compared to naive importance weighting (Precup, 2000). These methods reframe the learning objective as maximizing expected reward, using the primal-dual correspondence between value-function linear programming and distribution optimization (Nachum et al., 2019b; Nachum & Dai, 2020). DICE calculates the importance ratio using either a forward method that minimizes the residual error of the transposed Bellman equation (Zhang et al., 2020), or a backward method that optimizes the value function via duality (Nachum et al., 2019a). Some variations add a regularization to the objective function, yielding a closed-form solution for the importance ratio (Nachum et al., 2019b; Lee et al., 2021). Despite their ability to provide unbiased policy evaluation, DICE-style methods yield arbitrarily large importance ratios when the dataset lacks sufficient state-action space coverage, a challenge particularly acute in scarce data settings.

## 3 METHOD

### 3.1 PRELIMINARIES

We formulate reinforcement learning problem in the context of a Markov Decision Process (MDP) $\mathcal{M} = \langle \mathcal{S}, \mathcal{A}, T, r, \gamma, \rho_0 \rangle$, where $\mathcal{S}$ is the state space, $\mathcal{A}$ is the action space, $T : \mathcal{S} \times \mathcal{A} \times \mathcal{S} \to [0, 1]$

specifies the transition probability $T(s'|s,a)$, $r : \mathcal{S} \times \mathcal{A} \to \mathbb{R}$ is the reward function, $\gamma$ is the discount factor, and $\rho_0 : \mathcal{S} \to [0,1]$ is the initial state distribution. The policy $\pi : \mathcal{S} \times \mathcal{A} \to [0,1]$ maps from a state to a distribution over actions. Given a policy $\pi$, consider the trajectory $\tau = \{s_0, a_0, s_1, a_1, \dots\}$ sampled by $\pi$, i.e., $s_0 \sim \rho_0, a_t \sim \pi(\cdot|a_t), s_{t+1} \sim T(\cdot|s_t, a_t)$, the **stationary state-action distribution** is defined as $d^\pi(s,a) = (1-\gamma)\sum_{t=0}^\infty \gamma^t \Pr(s_t = s, a_t = a)$. The goal of RL is to learn a return-maximization policy $\pi^* = \arg\max_\pi \mathbb{E}_{\tau \sim \pi}[\sum_{t=0}^\infty \gamma^t r(s_t, a_t)]$, which is equivalent to reward maximization (Puterman, 2014): $\pi^* = \arg\max_\pi \mathbb{E}_{s,a \sim d^\pi}[r(s,a)]$.

In offline RL, the agent learns the policy from a pre-collected dataset $\mathcal{D} = \{(s_i, a_i, r_i, s_i')\}_{i=1}^N$. For simplicity, we denote the empirical state-action distribution of offline dataset as $d^\mathcal{D}$. The DICE-style methods apply marginal IS to estimate the expection of certain function $g$: $\mathbb{E}_{s,a \sim d^\pi}[g(s,a)] = \mathbb{E}_{s,a \sim d^\mathcal{D}}[\frac{d^\pi(s,a)}{d^\mathcal{D}(s,a)}g(s,a)]$ with $d^\pi, d^\mathcal{D}$ as target and proposal distributions. The IS estimation can thus be approximated by sampling from offline dataset.

## 3.2 CONSERVATIVE DENSITY ESTIMATION

In this section, we present Conservative Density Estimation (CDE), which aims to learns a policy that induces the distribution with conservative density in unseen state-action region. We first consider a $f$-divergence regularized policy optimization problem (Nachum et al., 2019a;b):

$$\max_{d^\pi \geq 0} \mathbb{E}_{d^\pi}[r(s,a)] - \alpha D_f(d^\pi \| d^\mathcal{D}), \quad s.t. \sum_a d^\pi(s,a) = (1-\gamma)\rho_0(s) + \gamma \mathcal{T}_* d^\pi(s), \forall s, \quad (1)$$

where $D_f(d^\pi \| d^\mathcal{D}) = \mathbb{E}_{d^\mathcal{D}}[f(\frac{d^\pi(s,a)}{d^\mathcal{D}(s,a)})]$ is the $f$-divergence between two distributions, $\alpha$ is the hyperparameter of regularization, $\mathcal{T}_* d^\pi(s) = \sum_{s',a'} T(s|s', a')d^\pi(s', a')$ is the transposed transition operator. Here we adhere to state-wise Bellman flow constraint as Lee et al. (2021) to incorporate the stochasticity of action distribution on next state $a' \sim \pi(\cdot|s')$ since the state-action-wise constraint can lead to overestimation for 'lucky' samples (Kostrikov et al., 2021b) and instability during training. Particularly, we have following assumption on $f$ function selection:

**Assumption 1.** *The f function in f-divergence is strictly convex and continuously differentiable, and* $(f')^{-1}(x) \geq 0, \forall x \in \mathbb{R}$.

The previous DICE methods (Nachum et al., 2019b; Nachum & Dai, 2020; Lee et al., 2021) transform constrained optimization problem to unconstrained one in Eq.(1) by Lagrange or Fenchel-Rockafellar duality and evaluate the unconstrained objective by marginal IS with $d^\mathcal{D}$ as proposal distribution. However, one **implicit assumption** behind DICE methods is that the support of dataset distribution is wide enough and otherwise the density of unseen state-actions in $d^\mathcal{D}$ can be zero or arbitrarily small. Therefore, when the support of $d^\pi$ mismatches $d^\mathcal{D}$, there will be a large extrapolation error for OOD state-actions and variance in IS estimation. Meanwhile, the $f$-divergence regularization is enforced on the support of data distribution and approximated by single or several sample points, failing to serve as an effective supervision to explicitly reduce extrapolation errors for unseen state-actions.

To overcome the above issues, we consider a new constraint on the density of $d^\pi(s,a)$ by $\mu(s,a)$: $d^\pi(s,a) \leq \epsilon\mu(s,a), \forall s, a \in \text{supp}(\mu)$, where $\mu(s,a)$ is a distribution on OOD state-action pairs, i.e., $\text{supp}(\mu) \cap \text{supp}(d^\mathcal{D}) = \emptyset$. The new optimization problem is formulated as

$$\max_{d^\pi \geq 0} \mathbb{E}_{d^\pi}[r(s,a)] - \alpha D_f(d^\pi \| d^\mathcal{D}) \quad (2)$$

$$s.t. \sum_a d^\pi(s,a) = (1-\gamma)\rho_0(s) + \mathcal{T}_* d^\pi(s), \forall s \quad (3)$$

$$d^\pi(s,a) \leq \epsilon\mu(s,a), \forall s, a \in \text{supp}(\mu). \quad (4)$$

The corresponding unconstrained problem is $\max_{d^\pi} \min_{\lambda \geq 0, v} \mathcal{L}(d^\pi, v, \lambda)$, where

$$\mathcal{L}(d^\pi, v, \lambda) = \mathbb{E}_{d^\pi}[A(s,a)] + (1-\gamma)\mathbb{E}_{\rho_0}[v(s_0)] - \alpha D_f(d^\pi \| d^\mathcal{D}) - \mathbb{E}_\mu[\lambda(s,a)(d^\pi/\mu(s,a) - \epsilon)] \quad (5)$$

and $A(s,a) := r(s,a) + \gamma\mathbb{E}_{s' \sim T(\cdot|s,a)}v(s') - v(s)$ is regarded as advantage function if we interpret $v(s)$ as the V-value of state $s$. The derivation is attached in Appendix A.1.

In practice, we restrict the state marginal of $\mu$ to match the state distribution of dataset $d^\mathcal{D}(s)$ as previous OOD querying methods (Kumar et al., 2020; Kostrikov et al., 2021a; Lyu et al., 2022) and

shrink the OOD region to unseen actions with existing states. Given a state $s$ in dataset, suppose there exists $n$ actions $a^{(1)}, \ldots, a^{(n)}$ such that $(s, a^{(i)}) \in \mathcal{D}, i = 1, \ldots, n$, we define the set of OOD actions as $\mathcal{A}_{\text{OOD}}(s) := \{a \mid \min_i \|a - a^{(i)}\|_\infty \geq \Delta a\}$. See more details in Appendix B.2.3. We further adopt a uniform distribution $\pi^\mu(a|s)$ over the unseen action space $\mathcal{A}_{\text{OOD}}(s)$ as the policy of $\mu$, i.e., $\mu(s, a) = d^\mathcal{D}(s)\pi^\mu(a|s)$. We want to emphasize that our method is also compatible with other OOD sampling distribution with proper inductive bias.

### 3.2.1 POLICY EVALUATION AND IMPROVEMENT

Based on the unconstrained objective in Eq.(5), we first adopt marginal IS to evaluate a policy given its stationary distribution. To avoid the support mismatch issue, we consider a new proposal distribution $\hat{d}^\mathcal{D}(s, a) := \zeta d^\mathcal{D}(s, a) + (1 - \zeta)\mu(s, a)$ in importance sampling, where $\zeta \in (0, 1)$ is the mixture coefficient. Therefore, the support of new proposal distribution can cover the target distribution $d^\pi$. We further replace the original $f$-divergence regularizer by $D_f(d^\pi \| \hat{d}^\mathcal{D})$ to constrain the density of both OOD and in-support state-actions. Besides, we substitute the importance ratio $w(s, a) = d^\pi(s, a)/\hat{d}^\mathcal{D}(s, a)$ for $d^\pi$ as $\hat{d}^\mathcal{D}$ is fixed. The new objective function is

$$\mathcal{L}'(w, v, \lambda) = \zeta \mathbb{E}_{d^\mathcal{D}}\left[w(s, a)A(s, a) - \alpha f(w(s, a))\right] + (1 - \gamma)\mathbb{E}_{\rho_0}[v(s_0)] \tag{6}$$
$$+ (1 - \zeta)\mathbb{E}_\mu\left[w(s, a)(A(s, a) - \lambda(s, a)) - \alpha f(w(s, a)) + \tilde{\epsilon}\lambda(s, a)\right],$$

where $\tilde{\epsilon} = \frac{\epsilon}{1-\zeta}$. The derivation is attached in Appendix A.1. With assumption 1, the objective in Eq.(2) is convex and thus is equivalent to the minimax problem $\min_{\lambda \geq 0, v} \max_{d^\pi} \mathcal{L}(d^\pi, v, \lambda)$ by Slater's condition. Moreover, the inner maximization has a closed-form solution (Nachum & Dai, 2020; Nachum et al., 2019b;a) and the outer minimization is a convex optimization problem. The proofs are in Appendix A.3, A.4.

**Proposition 1.** *With assumption 1, the closed-form solution to inner maximization problem* $\max_{w \geq 0} \mathcal{L}'(w, v, \lambda)$ *is*

$$w^*(s, a) = (f')^{-1}(\tilde{A}(s, a)/\alpha), \tag{7}$$

*where* $\tilde{A}(s, a) := A(s, a) - \mathbf{1}\{(s, a) \in supp(\mu)\} \cdot \lambda(s, a)$ *denotes **regularized advantage** function and* $\mathbf{1}\{\cdot\}$ *is the indicator function.*

**Proposition 2.** *The outer minimization problem* $\min_{\lambda \geq 0, v} \mathcal{L}'(w^*, v, \lambda)$ *is a convex optimization problem. Suppose the optimal solution is* $(\lambda^*, v^*)$, *then* $\lambda^*$ *has a closed-form solution*

$$\lambda^*(s, a) = \max\{0, A^*(s, a) - \alpha f'(\tilde{\epsilon})\}, \forall s, a \in supp(\mu), \tag{8}$$

*where* $A^*(s, a) = r(s, a) + \gamma \mathbb{E}_{s' \sim T(\cdot|s,a)} v^*(s') - v^*(s)$. *The optimal regularized advantage is*

$$\tilde{A}^*(s, a) = \begin{cases} A^*(s, a), & (s, a) \in supp(d^\mathcal{D}) \\ \min\{\alpha f'(\tilde{\epsilon}), A^*(s, a)\}, & (s, a) \in supp(\mu) \end{cases} \tag{9}$$

Based on the closed-form relation between stationary distribution $d^\pi$ and value function, we can thus improve the policy by maximizing w.r.t. value function. Since it requires the reward $r(s, a)$ and transition $T(\cdot|s, a)$ to compute regularized advantage function $\tilde{A}(s, a)$, which is available only for $(s, a) \in \mathcal{D}$, we consider function approximation for both V-value $v$ and regularized advantage $\tilde{A}$ by parameters $\varphi$ and $\phi$. The optimization is in two steps: 1) We first optimize $v_\varphi$ by minimizing the value of states in distribution:

$$\min_\varphi \mathbb{E}_{d^\mathcal{D}}\left[w^*(s, a)(r(s, a) + \gamma \mathbb{E}_{s'} v_\varphi(s') - v_\varphi(s)) - \alpha f(w^*(s, a))\right] + (1-\gamma)\mathbb{E}_{s_0 \sim \rho_0}[v_\varphi(s_0)]. \tag{10}$$

2) Then we regress the regularized advantage $\tilde{A}_\phi$ to the optimal $\tilde{A}^*$ in Eq.(9). Specifically, we regress the OOD advantages to $\alpha f'(\tilde{\epsilon})$ if they exceed it and regress the in-distribution advantages to the values from $v_\varphi$: $A_\varphi(s, a) = r(s, a) + \gamma \mathbb{E}_{s'} v_\varphi(s') - v_\varphi(s)$. In summary, we optimize the regularized advantage function by following mean squared error (MSE):

$$\min_\phi \zeta \mathbb{E}_{d^\mathcal{D}}[(\tilde{A}_\phi(s, a) - A_\varphi(s, a))^2] + (1 - \zeta)\mathbb{E}_\mu[\mathbf{1}\{\tilde{A}_\phi(s, a) > \alpha f'(\tilde{\epsilon})\}(\tilde{A}_\phi(s, a) - \alpha f'(\tilde{\epsilon}))^2], \tag{11}$$

and obtain the approximated optimal importance ratios for both in-distribution and OOD state-actions:

$$\tilde{w}^*(s, a) = (f')^{-1}(\tilde{A}_\phi(s, a)/\alpha). \tag{12}$$

By definition, the optimal distribution is $d^*(s, a) = \tilde{w}^*(s, a)\hat{d}^\mathcal{D}(s, a)$. In practice, we introduce another constraint to enforce $\sum_{s,a} d^*(s, a) = 1$ as (Zhang et al., 2020). See Appendix A.2 for full derivations.

### 3.2.2 POLICY EXTRACTION

Finally, we extract the policy from the learned importance ratios. Since one policy is uniquely determined given its corresponding stationary distribution, we extract the policy by minimizing the KL divergence between the stationary distributions of optimal policy and parameterized policy $\pi_\theta$. Meanwhile, as we have no access to $d^{\pi_\theta}(s, a)$ in offline setting, we estimate it by $d^*(s)\pi_\theta(a|s)$, where the optimal state marginal $d^*$ can be viewed as a state distribution of successful trajectories in dataset. The objective of policy extraction is

$$\min_\theta D_{\mathrm{KL}}[d^{\pi_\theta}\|d^*] \approx \min_\theta \mathbb{E}_{s\sim d^*, a\sim\pi_\theta} \left[ \log \frac{\hat{d}^{\mathcal{D}}(s, a)}{d^*(s, a)} + \log \frac{\pi_\theta(a|s)}{\hat{\pi}^{\mathcal{D}}(a|s)} + \log \frac{d^*(s)}{d^{\mathcal{D}}(s)} \right] \tag{13}$$

$$= \min_\theta \mathbb{E}_{s\sim d^*, a\sim\pi_\theta}[-\log \tilde{w}^*(s, a)] + \mathbb{E}_{s\sim d^*}[D_{\mathrm{KL}}[\pi_\theta(\cdot|s)\|\hat{\pi}^{\mathcal{D}}(\cdot|s)]], \tag{14}$$

where $\hat{\pi}^{\mathcal{D}}(a|s) = \zeta\pi^{\mathcal{D}}(a|s) + (1-\zeta)\pi^\mu(a|s)$ is the mixed behavior policy and $\pi^{\mathcal{D}}(a|s)$ denotes the empirical behavior policy. We will analyze the error induced by state marginal approximation in Theorem 2. The final objective in Eq.(14) consists of two components: the maximizing of $\tilde{w}^*$ and minimizing the divergence with mixed behavior policy, indicating the trade-off between performance improvement by maximizing the value and conservative learning to reduce extrapolation error.

The key steps of complete training procedure are summarized in Algo. 1. See Appendix B.2 for full algorithm and training details. One noteworthy difference from other actor-critic methods is that CDE updates the policy after the value function converges, which improves the learning stability and computation efficiency.

The advantages of CDE over previous DICE methods are two-fold: 1) the proposal distribution (i.e., $\hat{d}^{\mathcal{D}}$) has wider coverage than $d^{\mathcal{D}}$, which mitigates the support mismatch in importance sampling and prevents the arbitrarily large importance ratio; 2) CDE produces a conservative estimation of density in OOD region. Compared to previous conservative methods, CDE de-

---

**Algorithm 1** Conservative Density Estimation

Initialize value functions $v_\varphi, \tilde{A}_\phi$, behavior policy $\pi^{\mathcal{D}}$, policy $\pi_\theta$.
1: ▷ *policy evaluation and improvement*
2: **for** training iteration $i$ **do**
3:     Sample batch $\{(s_i, a_i, r_i, s_i')\}$ from $\mathcal{D}$ and $n$ OOD actions $\{a^{(1)}, \dots, a^{(n)}\}$ for each $s$;
4:     Update V-value $v_\varphi$ by Eq.(10);
5:     Update regularized advantage $\tilde{A}_\phi$ by Eq.(11);
6:     Update $\pi^{\mathcal{D}}$ by behavioral cloning.
7: **end for**
8: ▷ *policy extraction*
9: **for** training iteration $j$ **do**
10:     Update policy $\pi_\theta$ by Eq.(14).
11: **end for**

---

termines the degree of conservatism precisely by optimal $\lambda$ in Proposition 2, mitigating overly pessimistic estimation and loss of the generalization ability (Lyu et al., 2022). Furthermore, CDE disentangles two optimization steps, i.e., learning the value function by convex optimization and extracting the policy from the optimal importance ratio, thereby reducing the compounded error amplified by the interleaved optimization (Brandfonbrener et al., 2021).

### 3.3 THEORETICAL ANALYSIS

CDE adopts a proposal distribution with broader support in marginal IS and explicitly constrains the stationary distribution density of the OOD region, resulting in a theoretical bound for the importance ratio, also known as concentrability coefficient (Munos, 2007; Rashidinejad et al., 2021).

**Proposition 3** (Upper bound of concentrability ratio on OOD state-actions)**.** *With assumption 1, the theoretical optimal importance ratio is upper bounded by* $w^*(s, a) \le \tilde{\epsilon}, \forall (s, a) \in supp(\mu)$.

The proof of Proposition 3 is in Appendix A.5. It should be noted that an unbounded importance ratio can cause unstable training for importance-sampling-based methods (Shi et al., 2022). We further bound the function approximation $\tilde{w}^*$ in Eq.(12) with following continuity assumption:

**Assumption 2** (Lipschitz continuity of $A_\phi(s, a)$)**.** *There is a constant $L > 0$ such that*

$$|A_\phi(s, a) - A_\phi(s, a')| \le L \cdot \|a - a'\|_\infty, \quad \forall a, a' \in \mathcal{A}_{OOD}(s), \forall s \in \mathcal{D}.$$

**Theorem 1** (Upper bound of function approximated concentrability ratio)**.** *Suppose that 1) the action space is d-dim, i.e., $\mathcal{A} \subset \mathbb{R}^d$, 2) the diameter of $\mathcal{A}$ is $M$, i.e., $\|a_1 - a_2\|_\infty \le M, \forall a_1, a_2 \in \mathcal{A}$, and*

*3) there are at least $N$ OOD action samples from $\mu$ given any state $s \in \mathcal{D}$. When the continuity assumption 2 holds, with probability at least $1 - \delta, \delta > 0$, we have*

$$\tilde{w}^*(s,a) \leq (f')^{-1}\left(f'(\epsilon) + \frac{\xi}{\alpha} + \frac{L}{\alpha}\left(\Delta a^d + \frac{M^d}{N}\log\frac{1}{\delta}\right)^{1/d}\right), \quad \forall (s,a) \in supp(\mu) \qquad (15)$$

*where $\xi$ is the maximum residual error of OOD regression in Eq.(11), $\Delta a$ is the radius of in-distribution region as previously defined.*

The proof of Theorem 1 is in Appendix A.6. Proposition 3 and Theorem 1 show that CDE inherently bounds the OOD concentrability coefficient. This coefficient is frequently assumed to be bounded in the variance or performance analysis in both off-policy evaluation and offline RL domains (Rashidinejad et al., 2021; Ma et al., 2022; Zhan et al., 2022), as an unbounded concentrability coefficient can lead to instability during training. As such, the CDE framework shows promise as a potential tool for reducing variance or establishing performance lower bounds in future research.

Meanwhile, CDE evaluates the policy within the stationary distribution space, enabling the computation of performance differences between policies based on the discrepancies in their respective stationary distributions. Consequently, we can establish the following bound on the performance gap between the learned and optimal policies.

**Theorem 2** (The upper bound of performance gap). *Suppose the maximum reward is $R_{max} = \max_{s,a}\|r(s,a)\|$, let $V^\pi(\rho_0) := \mathbb{E}_{s_0 \sim \rho_0}[V^\pi(s_0)]$ denote the performance given a policy $\pi$. For policy $\pi$ optimized by Eq.(14) and $N$ transition data from $d^{\mathcal{D}}$, under mild assumptions, we have*

$$V^*(\rho_0) - V^\pi(\rho_0) \leq \frac{2R_{max}}{1-\gamma}D_{\mathrm{TV}}(d^{\mathcal{D}}(s)\|d^*(s)) + e_N \qquad (16)$$

*and $e_N$ converges in probability to zero at the rate $N^{-\frac{1}{4+h}}, \forall h > 0$, i.e., $N^{\frac{1}{4+h}}e_N \xrightarrow{N\to\infty} 0$ in probability. Here, $d^{\mathcal{D}}(s), d^*(s)$ denote the state marginal of $d^{\mathcal{D}}, d^*$, and $V^*(\rho_0)$ denotes the performance of optimal policy.*

The full assumptions and proof are in Appendix A.7. The performance gap bound comprises two elements: 1) the discrepancy between the state distribution of the data and the optimal policy, and 2) the number of training samples. The first element stems from the state-marginal approximation in Eq.(13) during policy extraction. Importantly, this bound explicitly highlights two **crucial factors** influencing the final performance of the learned policy: the performance of behavior policy $\pi^{\mathcal{D}}$ and the size of the offline dataset. It provides a quantitative illustration of how the offline RL problem difficulty increases as the performance of behavior policy degrades and the dataset size decreases.

## 4 EXPERIMENT

In this section, we aim to study if CDE can truly combine the advantages of both pessimism-based methods and the DICE-based approaches. We are particularly interested in two main questions:

(1) Does CDE incorporate the strengths of the stationary-distribution correction training framework when handling *sparse reward settings*?

(2) Can CDE's explicit density constraint effectively manage out-of-distribution (OOD) extrapolation issues in situations with *insufficient datasets*?

**Tasks**. To answer these questions, we adopt 3 Maze2D datasets, 8 Adroit datasets, and 6 MuJoCo (medium, medium-expert) datasets from the D4RL benchmark (Fu et al., 2020). The original rewards of Maze2D and Adroit tasks are sparse so we adopt the normalized score as the evaluation metric. Since the MuJoCo tasks are with dense rewards, we convert them to sparse-reward ones in the setting of goal reaching. Specifically, we first set the 75-percentile of all returns (the sum of rewards) in the dataset as return threshold. Subsequently, we assign a reward of 0 to all trajectories in the lower 75% of returns, and a reward of 1 to the top 25% of trajectories if the cumulative dense reward surpasses the threshold, which means the agent reaches the "goal". We compare the success rate of different methods on **sparse-MuJoCo** tasks. We adopt "-v1" tasks for Maze2D and Adroit domains and "-v2" tasks for MuJoCo domain. To assess the performance under scarce data conditions, we employ a random sampling strategy on the full dataset.

**Baselines**. We compare our CDE method with a collection of state-of-the-art offline RL baselines spanning different categories. These include: 1) behavior cloning (BC); 2) BCQ (Fujimoto et al., 2019) as a direct policy constraint method; 3) CQL (Kumar et al., 2020) as a value regularization method; 4) IQL (Kostrikov et al., 2021b) as an asymmetric Q-learning method; 5) TD3+BC (Fujimoto & Gu, 2021) as an implicit policy regularization method; 6) AlgaeDICE (Nachum et al., 2019b) as a policy-gradient-based DICE method; and 7) OptiDICE (Lee et al., 2021) as an in-sample DICE method. More details regarding the tasks and baselines are available in Appendix B.1.

**Training details**. We use tanh-squashed Gaussian policy for CDE's policy $\pi$ following SAC(Haarnoja et al., 2018) and tanh-squashed Gaussian mixture model for empirical behavior policy $\pi^{\mathcal{D}}$ to improve the expressivity for multi-modality of offline data from composite policies (e.g., medium-expert tasks in MuJoCo) or non-Markovian policies (e.g., Maze2D tasks). Following Lee et al. (2021), we adopt soft-chi function $f_{\text{soft}-\chi^2}$ in $f$-divergence and thus the $(f')^{-1}$ is equal to ELU function (Clevert et al., 2015) plus one, which satisfies Assumption 1 and also avoids the gradient vanishing problem for small values when computing importance ratios. For consistent evaluation and fair comparison, we keep hyperparameters the same for experiments in the same domain. We evaluate all methods every 1000 training steps and compute a mean value over 20 trajectories. The reported scores are the average of last 5 evaluation values with 5 seeds. We adopt the scores of baselines if they are reported in original paper. Full experimental details are included in Appendix B.2.

## 4.1 RESULTS ON D4RL SPARSE REWARD TASKS

Table 1 and 2 present the normalized scores and success rate, respectively. See more results in Appendix B.3. CDE consistently matches or surpasses the performance of the best baseline iacross nearly all tasks, achieving a particularly noteworthy margin of improvement in the Maze2D domain. On sparse-MuJoCo tasks, BCQ, CQL, and TD3+BC, while achieving high scores in dense-reward settings, display vulnerability to value function approximation errors due to reward sparsity, resulting in inferior performance. The substantial improvement CDE exhibits over over standard-RL-based methods highlights its capability to mitigate compounded value estimation error by leveraging a closed-form optimal value solution instead of Bellman bootstrapping value update.

Table 1: Normalized scores of CDE against other baselines on D4RL sparse-reward tasks. We **bold** the mean values that $\geq 0.99 *$ highest value.

| Task | BC | BCQ | CQL | IQL | TD3+BC | Algae-DICE | OptiDICE | CDE |
|---|---|---|---|---|---|---|---|---|
| maze2d-umaze | 3.8 | 32.8 | 5.7 | 50.0 | 41.5 | -15.7 | 111.0±8.3 | **134.1**±10.4 |
| maze2d-medium | 30.3 | 20.7 | 5.0 | 31.0 | 76.3 | 10.0 | 145.2±17.5 | **146.1**±13.1 |
| maze2d-large | 5.0 | 47.8 | 12.5 | 58.0 | 77.8 | -0.1 | 155.7±33.4 | **210.0**±13.5 |
| pen-human | 63.9 | 68.9 | 37.5 | 71.5 | 2.0 | -3.3 | 42.1±15.3 | **72.1**±15.8 |
| hammer-human | 1.2 | 0.5 | **4.4** | 1.4 | 1.4 | 0.3 | 0.3±0.0 | 1.9±0.7 |
| door-human | 2.0 | 0.0 | **9.9** | 4.3 | -0.3 | 0.0 | 0.1±0.1 | 7.7±3.3 |
| relocate-human | 0.1 | -0.1 | 0.2 | 0.1 | -0.3 | -0.1 | -0.1±0.1 | **0.3**±0.1 |
| pen-expert | 85.1 | **114.9** | 107.0 | 111.7 | 79.1 | -3.5 | 80.9±31.4 | 105.0±12.3 |
| hammer-expert | 125.6 | 107.2 | 86.7 | 116.3 | 3.1 | 0.3 | **127.0**±3.0 | **126.3**±3.4 |
| door-expert | 34.9 | 99.0 | 101.5 | 103.8 | -0.3 | 0.0 | 103.4±2.8 | **105.9**±0.3 |
| relocate-expert | 101.3 | 41.6 | 95.0 | **102.7** | -1.5 | -0.1 | 99.7±4.2 | **102.6**±1.9 |

Table 2: Success rate (%) of CDE against other baselines on sparse-MuJoCo tasks.

| Task | BCQ | CQL | IQL | TD3+BC | OptiDICE | CDE |
|---|---|---|---|---|---|---|
| halfcheetah-medium | 57.8±13.2 | **97.6**±4.1 | 76.6±5.8 | 41.6±17.6 | 80.0±3.4 | 82.0±8.6 |
| walker2d-medium | 41.0±11.5 | 17.7±10.4 | 19.5±4.2 | 21.0±16.7 | 38.4±13.3 | **53.0**±11.7 |
| hopper-medium | 2.0±4.0 | 74.0±5.0 | 0.0±0.0 | 0.0±0.0 | 81.0±5.7 | **85.5**±5.7 |
| halfcheetah-medium-expert | 24.8±9.8 | 4.2±5.8 | **95.4**±4.2 | 0.0±0.0 | 90.8±5.0 | **95.2**±2.9 |
| walker2d-medium-expert | 87.0±13.4 | 61.6±23.5 | 94.6±5.9 | 32.2±22.8 | 69.0±18.8 | **97.0**±2.8 |
| hopper-medium-expert | 20.0±11.0 | 0.0±0.0 | 94.8±2.8 | 22.0±10.8 | **97.4**±1.0 | **97.0**±1.4 |

Another notable observation is that CDE exceeds both AlgaeDICE and OptiDICE in most tasks. AlgaeDICE falls short because it updates the policy via high-variance policy gradients, as opposed

to extracting from optimal importance ratios. OptiDICE exhibits comparable performance in full data setting because the out-of-support issue is alleviated with adequate data coverage on state-action space. To further substantiate the efficacy of the constraints on unseen regions, we further investigate into situations where offline the data is scarce and the empirical state-action occupation is sparse.

## 4.2 COMPARATIVE EXPERIMENTS IN SCARCE DATA SETTING

In this section, we examine the performances across datasets of varying sizes. Given the extreme difficulty of the Adroit "-human" tasks due to their high-dimensional space, narrow data distribution, and data scarcity (25 trajectories), we select Maze2D and sparse-MuJoCo tasks as our testing platforms. We randomly sample 1%, 3%, 10%, and 30% of trajectories from standard datasets to create our sub-datasets. The evaluation process aligns with the full dataset experiments.

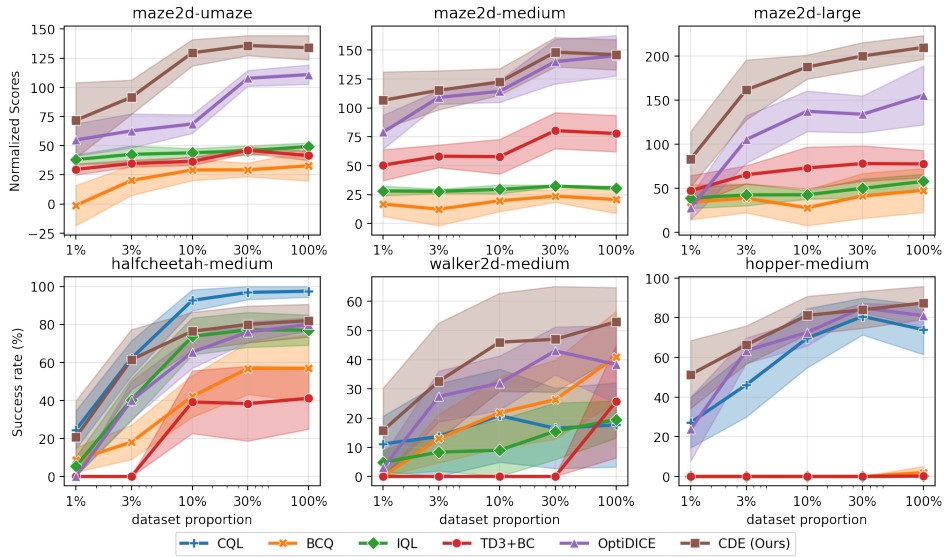

Figure 1: The results on sub-datasets with different dataset sizes.

The comparison results are shown in fig. 1, we defer the results on medium-expert MuJoCo tasks to Appendix B.3.3. CDE consistently achieves the highest scores across all dataset sizes in Maze2D domain. On the sparse-MuJoCo tasks, CDE outperforms most baselines, which may have high success rate with enough data, especially when data are scare (e.g., $\leq 3\%$). Notably, OptiDICE displays less robustness against scarce data, undergoing a sharp performance drop despite achieving comparable performances in full-data setting. Moreover, considering that the original Adroit "-human" tasks already contain scarce data, OptiDICE also falls short as shown in Table 1. This is because the narrow distribution of scarce data exacerbates the support mismatch problem in OptiDICE, leading to a significant bias in the importance-sampling-based off-policy evaluation. Conversely, CDE employs a mixed data policy, successfully mitigating large distribution shifts between the stationary distribution support of the dataset and the learned policy.

## 4.3 PARAMETER STUDIES

The results underscore the effectiveness of incorporating conservatism into density estimation in CDE. However, it prompts the question: given that conservative value function estimation is also prevalent in standard-RL-based offline RL methods (e.g., CQL), why do they underperform compared to CDE? To answer this question, we conduct a parameter study on $\tilde{\epsilon}$, which upper bounds the OOD importance ratio by Theorem 1, to delve deeper into the relationship between the performance of CDE and the degree of conservativeness as a parameter study. Besides, we also run the study on the distribution mixture coefficient $\zeta$. More results on parameter studies are attached in Appendix B.4.

**Max OOD IS ratio** $\tilde{\epsilon}$. We select the maze2d-large task where the agent's position directly represents the stationary distribution of the learned policy. We contrast CDE and CQL, two representative categories of pessimism augmentation in offline learning.

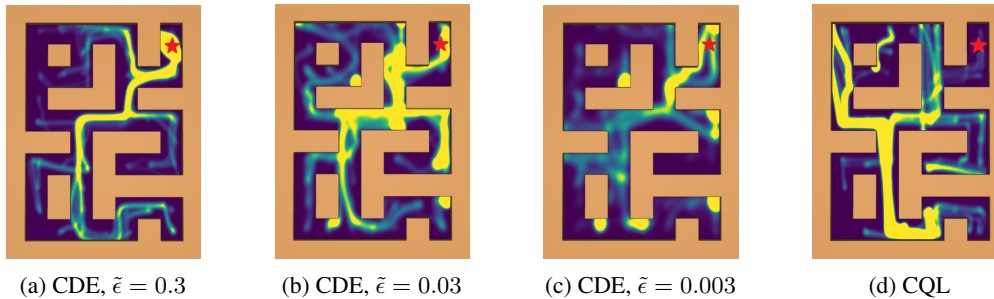

(a) CDE, $\tilde{\epsilon} = 0.3$      (b) CDE, $\tilde{\epsilon} = 0.03$      (c) CDE, $\tilde{\epsilon} = 0.003$      (d) CQL

Figure 2: The heatmaps of agents with different levels of conservatism in maze2d-large environment. Yellow denotes the high occupation probability. The starting point of each trajectory may vary but the destination (red star) is the same. Smaller $\tilde{\epsilon}$ indicates more conservative policy. The yellow accumulation points except the destination indicate that the agent is stuck at those regions.

The heatmap in fig.2 reveals that the probability mass at the midway and starting points increases as the level of conservatism escalates. In particular, in fig.2c, the agent is trapped at yellow points due to the overly strict constraint. Although stronger regularization can reduce OOD extrapolation error, excessive conservatism can harm generalization and lead to significant performance degradation. Compared to CDE, CQL policy is less likely to be trapped in single points but still struggles to reach the destination. There are two principal reasons: 1) the sparse reward setting poses significant challenges to the precise estimation of the value function; 2) CQL incorporates an additional minimization term on the unseen Q-value to induce pessimism and balances it with a penalty coefficient, which lacks a theoretical foundation on the conservatism adjustment. In contrast, the CDE employs the closed-form relation between value and density to explicitly constrain the stationary distribution space, allowing for more precise control over the level of conservatism.

**Mixture coefficient** $\zeta$. We test the performance on sparse-MuJoCo tasks with varying $\zeta$. Fig. 3 shows that the performance is not very sensitive to $\zeta$ when it is closed to 1 but the success rate decreases when $\zeta$ is smaller. This is because 1) the $f$-divergence regularizer enforces learned policy to remain close to mixture behavior policy, which tends to take unseen actions more frequently when $\zeta$ decreases; and 2) a lower $\zeta$ can cause the objective in eq.(11) to be dominated by OOD learning, overshadowing in-distribution learning. Therefore, we choose $\zeta = 0.9$ for all tasks in practice.

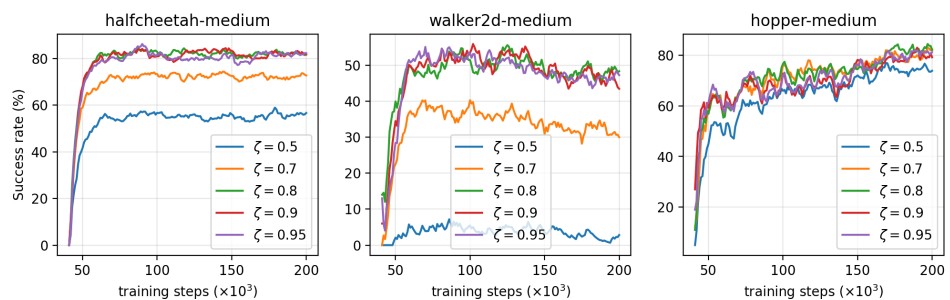

Figure 3: The performances with different $\zeta$.

## 5 CONCLUSION

In this work, we propose CDE, a new offline RL approach, derived from the perspective of stationary state-action occupation. CDE applies the pessimism mechanism on stationary distribution and enjoys the benefits from both fields, which makes it perform advantageously in sparse reward and scarce data settings. We further provide the theoretical analysis for the importance-sampling ratios and performance of CDE. Extensive experimental results demonstrated remarkable improvements over previous baselines in challenging tasks, highlighting its practical potential for real-world applications.

The major limitation of CDE is that it requires the strict alignment of initial state distribution in offline data and online environments, which restricts its performance in inconsistent settings. To overcome this drawback, one future direction can be extending the current framework to goal-conditioned RL setting.

ACKNOWLEDGMENTS

The research is partly supported by the National Science Foundation under grants CNS-2047454.

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

## A  SUPPLEMENTARY DERIVATIONS AND PROOFS

### A.1  DERIVATION OF EQ.(5)(6)

**Derivations of Eq.(5) and Slater's Condition**.

Given the optimization problem:

$$\max_{d^\pi \geq 0} \mathbb{E}_{d^\pi}[r(s,a)] - \alpha D_f(d^\pi \| d^{\mathcal{D}})$$

$$s.t. \sum_a d^\pi(s,a) = (1-\gamma)\rho_0(s) + \mathcal{T}_* d^\pi(s), \forall s$$

$$d^\pi(s,a) \leq \epsilon \mu(s,a), \forall s, a \in \text{supp}(\mu).$$

The corresponding unconstrained problem is

$$\max_{d^\pi} \min_{\lambda \geq 0, v} \mathcal{L}(d^\pi, v, \lambda) \mathcal{L}(d^\pi, v, \lambda) = \mathbb{E}_{d^\pi}[A(s,a)] + (1-\gamma)\mathbb{E}_{\rho_0}[v(s_0)] - \alpha D_f(d^\pi \| d^{\mathcal{D}})$$

$$- \mathbb{E}_\mu \left[ \lambda(s,a) \left( d^\pi / \mu(s,a) - \epsilon \right) \right]$$

where $A(s,a) := r(s,a) + \gamma \mathbb{E}_{s' \sim T(\cdot|s,a)} v(s') - v(s)$.

*Proof.* The Lagrangian for constrained optimization is

$$\max_{d^\pi} \min_{\lambda \geq 0, v} \mathcal{L}(d^\pi, v, \lambda) := \mathbb{E}_{\substack{(s,a) \sim d^\pi \\ s' \sim T(\cdot|s,a)}} [r(s,a)] - \sum_{s,a \in \text{supp}(\mu)} \lambda(s,a)[d^\pi(s,a) - \epsilon\mu(s,a)] +$$

$$\sum_s v(s)[(1-\gamma)\rho_0(s) + \gamma \mathcal{T}_* d^\pi(s) - \sum_a d^\pi(s,a)] - \alpha D_f(d^\pi \| d^{\mathcal{D}}) \tag{17}$$

$$= \mathbb{E}_{d^\pi}[r(s,a)] - \mathbb{E}_\mu \left[ \lambda(s,a) \left( \frac{d^\pi}{\mu} - \epsilon \right) \right] - \alpha D_f(d^\pi \| d^{\mathcal{D}})$$

$$+ (1-\gamma)\mathbb{E}_{\rho_0}[v(s_0)] + \sum_s v(s) \sum_{\bar{s}, \bar{a}} T(s|\bar{s}, \bar{a}) d^\pi(\bar{s}, \bar{a}) - \mathbb{E}_{d^\pi}[v(s)] \tag{18}$$

$$= \mathbb{E}_{d^\pi}[r(s,a)] - \mathbb{E}_\mu \left[ \lambda(s,a) \left( \frac{d^\pi}{\mu} - \epsilon \right) \right] - \alpha D_f(d^\pi \| d^{\mathcal{D}})$$

$$+ (1-\gamma)\mathbb{E}_{\rho_0}[v(s_0)] + \sum_{s'} v(s') \sum_{s,a} T(s'|s,a) d^\pi(s,a) - \mathbb{E}_{d^\pi}[v(s)] \tag{19}$$

$$= \mathbb{E}_{d^\pi}[r(s,a)] - \mathbb{E}_\mu \left[ \lambda(s,a) \left( \frac{d^\pi}{\mu} - \epsilon \right) \right] - \alpha D_f(d^\pi \| d^{\mathcal{D}})$$

$$+ (1-\gamma)\mathbb{E}_{\rho_0}[v(s_0)] + \mathbb{E}_{s,a \sim d^\pi, s' \sim T(\cdot|s,a)}[v(s')] - \mathbb{E}_{d^\pi}[v(s)] \tag{20}$$

$$= \mathbb{E}_{d^\pi} \left[ r(s,a) + \mathbb{E}_{s' \sim T(\cdot|s,a)}[v(s')] - v(s) \right] - \mathbb{E}_\mu \left[ \lambda(s,a) \left( \frac{d^\pi}{\mu} - \epsilon \right) \right]$$

$$- \alpha D_f(d^\pi \| d^{\mathcal{D}}) + (1-\gamma)\mathbb{E}_{\rho_0}[v(s_0)] \tag{21}$$

$$= \mathbb{E}_{d^\pi}[A(s,a)] + (1-\gamma)\mathbb{E}_{\rho_0}[v(s_0)] - \alpha D_f(d^\pi \| d^{\mathcal{D}}) - \mathbb{E}_\mu \left[ \lambda(s,a) \left( \frac{d^\pi}{\mu} - \epsilon \right) \right] \tag{22}$$

$$\square$$

**The Slater's Condition for Problem in Eq.(2-4)**.

The corresponding Slater's condition for optimization problem in eq.(2-4) is that there exists a strictly feasible state-action distribution $d(s,a)$ s.t. the equality constraint in eq.(3) holds while constraint in eq.(4) is satisfied with strict inequality. One strictly feasible solution is $d^{\pi_{\mathcal{D}}}(s,a)$, the stationary state-action distribution of behavior policy $\pi_{\mathcal{D}}$:

1. Bellman flow constraint $\sum_a d^{\pi_{\mathcal{D}}}(s,a) = (1-\gamma)\rho_0(s) + \mathcal{T}_* d^{\pi_{\mathcal{D}}}(s)$ holds because the $\pi_{\mathcal{D}}$ is a valid policy and $d^{\pi_{\mathcal{D}}}(s,a)$ is its corresponding stationary distribution;

2. $d^{\pi_\mathcal{D}}(s,a) = 0 < \epsilon\mu(s,a), \forall s,a \in \mathrm{supp}(\mu)$ by definition of $\mu$.

Therefore, the Slater's condition holds.

**Derivations of Eq.(6)**.

Replace $D_f(d^\pi\|d^\mathcal{D})$ by $D_f(d^\pi\|\hat{d}^\mathcal{D})$ as new regularizer, and use $\hat{d}^\mathcal{D}(s,a) := \zeta d^\mathcal{D}(s,a) + (1-\zeta)\mu(s,a)$ as proposal distribution, we can obtain

$$\mathcal{L}'(w,v,\lambda) = \zeta\mathbb{E}_{d^\mathcal{D}}\left[w(s,a)A(s,a) - \alpha f(w(s,a))\right] + (1-\gamma)\mathbb{E}_{\rho_0}[v(s_0)]$$
$$+ (1-\zeta)\mathbb{E}_\mu\left[w(s,a)(A(s,a) - \lambda(s,a)) - \alpha f(w(s,a)) + \tilde{\epsilon}\lambda(s,a)\right].$$

*Proof.*

$$\mathcal{L}'(w,v,\lambda) = \mathbb{E}_{d^\pi}[A(s,a)] + (1-\gamma)\mathbb{E}_{\rho_0}[v(s_0)] - \alpha D_f(d^\pi\|\hat{d}^\mathcal{D}) - \mathbb{E}_\mu\left[\lambda(s,a)\left(\frac{d^\pi}{\mu} - \epsilon\right)\right] \tag{23}$$

$$= \mathbb{E}_{\hat{d}^\mathcal{D}}\left[\frac{d^\pi}{\hat{d}^\mathcal{D}}A(s,a) - \alpha f\left(\frac{d^\pi}{\hat{d}^\mathcal{D}}\right)\right] + (1-\gamma)\mathbb{E}_{\rho_0}[v(s_0)] - \mathbb{E}_\mu\left[\lambda(s,a)\left(\frac{d^\pi}{\mu} - \epsilon\right)\right] \tag{24}$$

$$= \mathbb{E}_{\hat{d}^\mathcal{D}}\left[w(s,a)A(s,a) - \alpha f(w(s,a))\right] + (1-\gamma)\mathbb{E}_{\rho_0}[v(s_0)] - \mathbb{E}_\mu\left[\lambda(s,a)\left(\frac{d^\pi}{\mu} - \epsilon\right)\right] \tag{25}$$

$$= \zeta\mathbb{E}_{d^\mathcal{D}}\left[w(s,a)A(s,a) - \alpha f(w(s,a))\right] + (1-\gamma)\mathbb{E}_{\rho_0}[v(s_0)]$$
$$+ (1-\zeta)\mathbb{E}_\mu\left[w(s,a)(A(s,a) - \lambda(s,a)) - \alpha f(w(s,a)) + \tilde{\epsilon}\lambda(s,a)\right], \tag{26}$$
$\square$

## A.2 Derivation of normalization for stationary distribution

In practice, the optimal distribution $d^*$ may not satisfy $\sum_{s,a} d^*(s,a) = 1$ due to function approximation error. Therefore, we explicitly enforce the $\sum_{s,a} d^*(s,a) = 1$ (Zhang et al., 2020) to make $d^*$ a valid distribution, which is equivalent to $\mathbb{E}_{\hat{d}^\mathcal{D}} w^*(s,a) = 1$.

With new normalization constraint, the corresponding unconstrained problem becomes

$$\min_{\lambda \geq 0, v, \eta} \max_w \mathcal{L}(w; v, \lambda, \eta) := \zeta\mathbb{E}_{d^\mathcal{D}}\left[w(s,a)A(s,a) - \alpha f(w(s,a))\right] + (1-\gamma)\mathbb{E}_{\rho_0}[v(s_0)]$$
$$+ (1-\zeta)\mathbb{E}_\mu\left[w(s,a)(A(s,a) - \lambda(s,a)) - \alpha f(w(s,a)) + \tilde{\epsilon}\lambda(s,a)\right] + \eta(1 - \mathbb{E}_{\hat{d}^\mathcal{D}} w^*(s,a)) \tag{27}$$

$$= \zeta\mathbb{E}_{d^\mathcal{D}}\left[w(s,a)(A(s,a) - \eta) - \alpha f(w(s,a))\right] + (1-\gamma)\mathbb{E}_{\rho_0}[v(s_0)]$$
$$+ (1-\zeta)\mathbb{E}_\mu\left[w(s,a)(A(s,a) - \lambda(s,a) - \eta) - \alpha f(w(s,a)) + \tilde{\epsilon}\lambda(s,a)\right] + \eta, \tag{28}$$

where $\eta$ is the dual variable of the normalization constraint. Therefore, we only need to replace $\tilde{A}$ by $\tilde{A} - \eta$ for optimization w.r.t. $v_\varphi$ and $\pi_\theta$. Meanwhile, we will also update $\eta$ by gradient descent. See more details in full algorithm in Appendix B.2.5.

## A.3 Proof for Proposition 1

With assumption 1, the closed-form solution to inner maximization problem $\max_{w \geq 0} \mathcal{L}'(w,v,\lambda)$ is

$$w^*(s,a) = (f')^{-1}(\tilde{A}(s,a)/\alpha),$$

where $\tilde{A}(s,a) := A(s,a) - \mathbf{1}\{(s,a) \in \mathrm{supp}(\mu)\} \cdot \lambda(s,a)$ denotes **regularized advantage** function and $\mathbf{1}\{\cdot\}$ is the indicator function.

*Proof.* Let $\frac{\partial \mathcal{L}'(w,v,\lambda)}{\partial w} = 0$ and we have

$$\zeta\mathbb{E}_{d^\mathcal{D}}[A(s,a) - \alpha f'(w(s,a))] + (1-\zeta)\mathbb{E}_\mu[A(s,a) - \lambda(s,a) - \alpha f'(w(s,a))] = 0 \tag{29}$$

Separate state-action space $\mathcal{S} \times \mathcal{A}$ into the support of $d^\mathcal{D}$ and $\mu$, then we can get the solution as Eq.(29). Meanwhile, $w^* \geq 0$ always holds by assumption 1. Therefore, the solution is valid and is exactly the optimal solution. $\square$

The closed-form solution to optimal importance ratio can also be derived by Fenchel-Rockafellar dual form of $f$-divergence (Nachum et al., 2019b; Nachum & Dai, 2020), which leads to the same results.

### A.4    PROOF FOR PROPOSITION 2

The outer minimization problem $\min_{\lambda \geq 0, v} \mathcal{L}'(w^*, v, \lambda)$ is a convex optimization problem. Suppose the optimal solution is $(\lambda^*, v^*)$, then $\lambda^*$ has a closed-form solution

$$\lambda^*(s, a) = \max\{0, A^*(s, a) - \alpha f'(\tilde{\epsilon})\}, \forall s, a \in \text{supp}(\mu),$$

where $A^*(s, a) = r(s, a) + \gamma \mathbb{E}_{s' \sim T(\cdot | s, a)} v^*(s') - v^*(s)$. The optimal regularized advantage is

$$\tilde{A}^*(s, a) = \begin{cases} A^*(s, a), & (s, a) \in \text{supp}(d^{\mathcal{D}}) \\ \min\{\alpha f'(\tilde{\epsilon}), A^*(s, a)\}, & (s, a) \in \text{supp}(\mu) \end{cases}$$

*Proof.* Notice that the convexity of dual function, which corresponds to $g(v, \lambda) := \max_w \mathcal{L}'(w, v, \lambda)$ in our setting, is proved by previous literature (Proposition 1, section 8.3 in (Luenberger, 1997)).

Then we prove the closed form of optimal $\lambda^*$. Consider the partial differential of $\mathcal{L}'(w^*, v, \lambda)$ w.r.t $\lambda$:

$$\frac{\partial \mathcal{L}'(w^*, v, \lambda)}{\partial \lambda} = \frac{\partial}{\partial \lambda} (1 - \zeta) \mathbb{E}_\mu \left[ (f')^{-1} \left( \frac{A - \lambda}{\alpha} \right) (A - \lambda) - \alpha f \left( (f')^{-1} \left( \frac{A - \lambda}{\alpha} \right) \right) + \lambda \tilde{\epsilon} \right] \tag{30}$$

$$= (1 - \zeta) \mathbb{E}_\mu \left[ ((f')^{-1})' \left( \frac{A - \lambda}{\alpha} \right) \left( -\frac{1}{\alpha} \right) (A - \lambda) - (f')^{-1} \left( \frac{A - \lambda}{\alpha} \right) \right.$$
$$\left. - \alpha f' \left( (f')^{-1} \left( \frac{A - \lambda}{\alpha} \right) \right) ((f')^{-1})' \left( \frac{A - \lambda}{\alpha} \right) \left( -\frac{1}{\alpha} \right) + \tilde{\epsilon} \right] \tag{31}$$

$$= (1 - \zeta) \mathbb{E}_\mu \left[ -((f')^{-1})' \left( \frac{A - \lambda}{\alpha} \right) \frac{A - \lambda}{\alpha} - (f')^{-1} \left( \frac{A - \lambda}{\alpha} \right) \right.$$
$$\left. + ((f')^{-1})' \left( \frac{A - \lambda}{\alpha} \right) \frac{A - \lambda}{\alpha} + \tilde{\epsilon} \right] \tag{32}$$

$$= (1 - \zeta) \mathbb{E}_\mu \left[ -(f')^{-1} \left( \frac{A - \lambda}{\alpha} \right) + \tilde{\epsilon} \right] \tag{33}$$

We omit $(s, a)$ for $A$ and $\lambda$ functions for brevity. By assumption 1, $f$ is convex and $f'$ is monotonic increasing. Therefore, when $A(s, a) \leq \alpha f'(\tilde{\epsilon})$, the gradient of $\lambda$ is always non-negative for $\lambda \geq 0$; otherwise, the gradient equals to zero when $\lambda = A(s, a) - \alpha f'(\tilde{\epsilon})$.

Therefore, the optimal solution of $\lambda$ is

$$\lambda^*(s, a) = \max\{0, A(s, a) - \alpha f'(\tilde{\epsilon})\}. \tag{34}$$

Plug-in the $\lambda^*$ to Proposition 1 and then we can get the optimal regularized advantage function $\tilde{A}^*$. $\qquad \square$

### A.5    PROOF FOR PROPOSITION 3

With assumption 1, the theoretical optimal importance ratio is upper bounded by $w^*(s, a) \leq \tilde{\epsilon}, \forall (s, a) \in \text{supp}(\mu)$.

*Proof.* Combine equation 7 and equation 8,

$$w^*(s, a) := (f')^{-1} \left( \frac{A(s, a) - \lambda^*(s, a)}{\alpha} \right) \tag{35}$$

$$= (f')^{-1} \left( \frac{A(s, a) - \max\{0, A(s, a) - \alpha f'(\tilde{\epsilon})\}}{\alpha} \right) \tag{36}$$

$$= (f')^{-1} \left( \min \left\{ \frac{A(s, a)}{\alpha}, f'(\tilde{\epsilon}) \right\} \right) \tag{37}$$

By Assumption 1, $f'$ is strictly increasing and so is $(f')^{-1}$. As a result,

$$w^*(s,a) = (f')^{-1}\left(\min\left\{\frac{A(s,a)}{\alpha}, f'(\tilde{\epsilon})\right\}\right) = \min\{(f')^{-1}(A(s,a)/\alpha), \tilde{\epsilon}\} \tag{38}$$

$$\square$$

### A.6 PROOF FOR THEOREM 1

We first give the following lemma.

**Lemma 1.** *Suppose that 1) the action space is $d$-dim, i.e., $\mathcal{A} \subset \mathbb{R}^d$, 2) the diameter of $\mathcal{A}$ is $M$, i.e., $\|a_1 - a_2\|_\infty \leq M, \forall a_1, a_2 \in \mathcal{A}$, and 3) there are $N$ action samples from $\mu$ given any state $s \in \mathcal{D}$, denoted by $(s, a_1), \dots, (s, a_N)$, and $\mu$ is a uniform distribution over OOD action space. Let $\delta > 0$, $(s, a) \in \mathcal{D}$, $\tilde{a} \in \mathcal{A}_{OOD}(s)$. We have*

$$\mathbb{P}\left(\min_{i=1,\dots,N}\|\tilde{a} - a_i\|_\infty > \delta\right) \leq \left(1 - \frac{\delta^d - \Delta a^d}{M^d}\right)^N \tag{39}$$

Now we consider the theorem:

Suppose that 1) the action space is $d$-dim, i.e., $\mathcal{A} \subset \mathbb{R}^d$, 2) the diameter of $\mathcal{A}$ is $M$, i.e., $\|a_1 - a_2\|_\infty \leq M, \forall a_1, a_2 \in \mathcal{A}$, and 3) there are at least $N$ OOD action samples from $\mu$ given any state $s \in \mathcal{D}$. When the continuity assumption 2 holds, with probability at least $1 - \delta, \delta > 0$, we have

$$\tilde{w}^*(s,a) \leq (f')^{-1}\left(f'(\epsilon) + \frac{\xi}{\alpha} + \frac{L}{\alpha}\left(\Delta a^d + \frac{M^d}{N}\log\frac{1}{\delta}\right)^{1/d}\right), \quad \forall(s,a) \in \text{supp}(\mu)$$

where $\xi$ is the maximum residual error of OOD regression in Eq.(11), $\Delta a$ is the radius of in-distribution region as previously defined.

*Proof.* Let $B_\infty(x, y) = \{x' \in \mathbb{R}^d : \|x - x'\|_\infty \leq y\}$ denote the $d$-dim Euclidean Ball under $\|\cdot\|_\infty$. The volume of $B_\infty(x, y)$ is then given by $\text{Vol}(B_\infty(x, y)) = 2^d y^d$. We have

$$\mathbb{P}(\|\tilde{a} - a_1\|_\infty > \delta) = 1 - \mathbb{P}(\|\tilde{a} - a_1\|_\infty \leq \delta) = 1 - \mathbb{P}(a_1 \in B_\infty(\tilde{a}, \delta)) \tag{40}$$

Recall that $(s, a_1), \dots, (s, a_N)$ are $i.i.d.$ samples from uniform distribution on $\mathcal{A} \backslash B_\infty(a, \Delta a)$. Thus, we can establish the following equality

$$\mathbb{P}(a_1 \in B_\infty(\tilde{a}, \delta)) = \int_{\mathbb{R}^d} \mathbf{1}\{x \in B_\infty(\tilde{a}, \delta)\}\mu(x)dx \tag{41}$$

$$= \int_{\mathbb{R}^d} \mathbf{1}\{x \in B_\infty(\tilde{a}, \delta)\}\frac{\mathbf{1}\{x \in \mathcal{A} \backslash B_\infty(a, \Delta a)\}}{\text{Vol}(\mathcal{A} \backslash B_\infty(a, \Delta a))}dx \tag{42}$$

$$= \frac{1}{\text{Vol}(\mathcal{A} \backslash B_\infty(a, \Delta a))}\int_{\mathbb{R}^d} \mathbf{1}\{x \in B_\infty(\tilde{a}, \delta) \cap \mathcal{A} \backslash B_\infty(a, \Delta a)\}dx \tag{43}$$

$$= \frac{\text{Vol}(B_\infty(\tilde{a}, \delta) \cap \mathcal{A} \backslash B_\infty(a, \Delta a))}{\text{Vol}(\mathcal{A} \backslash B_\infty(a, \Delta a))} \tag{44}$$

Since the action space $\mathcal{A}$ is bounded with radius $M$,

$$\text{Vol}(\mathcal{A} \backslash B_\infty(a, \Delta a)) \leq \text{Vol}(B_\infty(a, M)) \tag{45}$$

In addition, notice that

$$\text{Vol}(B_\infty(\tilde{a}, \delta) \cap \mathcal{A} \backslash B_\infty(a, \Delta a)) \geq \text{Vol}(B_\infty(\tilde{a}, \delta)) - \text{Vol}(B_\infty(a, \Delta a)) \tag{46}$$

Combining the above inequalities and plugging in the formula for $d$-dim ball under $\|\cdot\|_\infty$, we have

$$\mathbb{P}(\|\tilde{a} - a_1\|_\infty > \delta) \tag{47}$$

$$= 1 - \mathbb{P}(a_1 \in B_\infty(\tilde{a}, \delta)) \tag{48}$$

$$\leq 1 - \frac{\text{Vol}(B_\infty(\tilde{a}, \delta)) - \text{Vol}(B_\infty(a, \Delta a))}{\text{Vol}(B_\infty(a, M))} \tag{49}$$

$$= 1 - \frac{\delta^d - \Delta a^d}{M^d} \tag{50}$$

By independence between the OOD samples

$$\mathbb{P}\left(\min_{i=1,\ldots,N} \|\tilde{a} - a_i\|_\infty > \delta\right) \tag{51}$$

$$=\mathbb{P}\left(\bigcap_{i=1}^{N}\{\|\tilde{a} - a_i\|_\infty > \delta\}\right) = \mathbb{P}(\|\tilde{a} - a_1\|_\infty > \delta)^N \le \left(1 - \frac{\delta^d - \Delta a^d}{M^d}\right)^N \tag{52}$$

This finishes the proof. □

As a remark, if we consider $\|\cdot\|_p$ instead of $\|\cdot\|_\infty$, the result would still be the same. Now we give the proof of Theorem 1.

*Proof.* Let $(s,a) \in \mathcal{D}$ and suppose that $(s,a_1),\ldots,(s,a_N)$ are the *i.i.d.* samples from $\mu$. Let $a' \in \{a_1,\ldots,a_N\}$ be the OOD sample that is closest to $a$ under $\|\cdot\|_\infty$ (i.e., $a' = \arg\min_{x\in\{a_1,\ldots,a_N\}} \|x - a\|_\infty$). Since the maximum regression residual error is $\xi$, we have

$$\tilde{A}_\phi(s, a') \le \alpha f'(\tilde{\epsilon}) + \xi. \tag{53}$$

Then, by assumption 2, we have

$$\tilde{A}_\phi(s,a) \le \tilde{A}_\phi(s,a') + |\tilde{A}_\phi(s,a) - \tilde{A}_\phi(s,a')| \le \alpha f'(\tilde{\epsilon}) + L \cdot \|a - a'\|_\infty + \xi \tag{54}$$

Let $\tilde{\delta} > 0$ and $\delta' = \frac{\alpha}{L}(f'(\tilde{\epsilon} + \tilde{\delta}) - f'(\tilde{\epsilon}) - \frac{\xi}{\alpha})$. Suppose $\delta' > 0$, by Lemma 1, and using the fact that $1 + x \le e^x$, $\forall x \in \mathbb{R}$, we have

$$\mathbb{P}(\|a' - a\|_\infty \le \delta') \ge 1 - \left(1 - \frac{\delta'^d - \Delta a^d}{M^d}\right)^N \ge 1 - e^{-N\frac{\delta'^d - \Delta a^d}{M^d}} \tag{55}$$

Combine equation 54 and equation 55, we have, with probability at least $1 - e^{-N\frac{\delta'^d - \Delta a^d}{M^d}}$,

$$\tilde{A}_\phi(s,a) \le \alpha f'(\tilde{\epsilon}) + L\delta' + \xi = \alpha f'(\tilde{\epsilon} + \tilde{\delta}) \tag{56}$$

Recall that $\tilde{w}^*(s,a) := (f')^{-1}(\tilde{A}_\phi(s,a)/\alpha)$. By equation 56, we have

$$\tilde{w}^*(s,a) := (f')^{-1}(\tilde{A}_\phi(s,a)/\alpha) \le (f')^{-1}(f'(\tilde{\epsilon} + \tilde{\delta})) = \tilde{\epsilon} + \tilde{\delta} \tag{57}$$

with probability at least $1 - e^{-N\frac{\delta'^d - \Delta a^d}{M^d}}$, where $\delta' = \frac{\alpha}{L}(f'(\tilde{\epsilon} + \tilde{\delta}) - f'(\tilde{\epsilon}) - \frac{\xi}{\alpha})$. The inequality step in equation 57 follows from the fact that $f'$ is increasing.

Let $\delta \in (0,1)$. Consider $\tilde{\delta} = (f')^{-1}(f'(\tilde{\epsilon}) + \frac{\xi}{\alpha} + \frac{L}{\alpha}(\Delta a^d + \frac{M^d}{N}\log\frac{1}{\delta})^{\frac{1}{d}}) - \tilde{\epsilon}$. First, we verify $\delta' > 0$ with this choice of $\tilde{\delta}$.

$$\delta' := \frac{\alpha}{L}\left(f'(\tilde{\epsilon} + \tilde{\delta}) - f'(\tilde{\epsilon}) - \frac{\xi}{\alpha}\right) \tag{58}$$

$$= \frac{\alpha}{L}\left(f'(\tilde{\epsilon}) + \frac{\xi}{\alpha} + \frac{L}{\alpha}\left(\Delta a^d + \frac{M^d}{N}\log\frac{1}{\delta}\right)^{\frac{1}{d}} - f'(\tilde{\epsilon}) - \frac{\xi}{\alpha}\right) \tag{59}$$

$$= \left(\Delta a^d + \frac{M^d}{N}\log\frac{1}{\delta}\right)^{\frac{1}{d}} \tag{60}$$

$$> 0 \tag{61}$$

Substitute $\tilde{\delta}$ back into equation 57. We get

$$w^*(s,a) \le \tilde{\epsilon} + (f')^{-1}\left(f'(\tilde{\epsilon}) + \frac{\xi}{\alpha} + \frac{L}{\alpha}\left(\Delta a^d + \frac{M^d}{N}\log\frac{1}{\delta}\right)^{\frac{1}{d}}\right) - \tilde{\epsilon} \tag{62}$$

$$= (f')^{-1}\left(f'(\tilde{\epsilon}) + \frac{\xi}{\alpha} + \frac{L}{\alpha}\left(\Delta a^d + \frac{M^d}{N}\log\frac{1}{\delta}\right)^{\frac{1}{d}}\right) \tag{63}$$

with probability of at least

$$1 - e^{-N \frac{\delta'^d - \Delta a^d}{M^d}} \tag{64}$$

$$= 1 - \exp\left( -N \frac{(\frac{\alpha}{L}(f'(\tilde{\epsilon} + \tilde{\delta}) - f'(\tilde{\epsilon}) - \frac{\xi}{\alpha}))^d - \Delta a^d}{M^d} \right) \tag{65}$$

$$= 1 - \exp\left( -N \frac{(\frac{\alpha}{L}(f'(\tilde{\epsilon}) + \frac{\xi}{\alpha} + \frac{L}{\alpha}(\Delta a^d + \frac{M^d}{N} \log \frac{1}{\delta})^{\frac{1}{d}} - f'(\tilde{\epsilon}) - \frac{\xi}{\alpha}))^d - \Delta a^d}{M^d} \right) \tag{66}$$

$$= 1 - \exp\left( -N \frac{\Delta a^d + \frac{M^d}{N} \log \frac{1}{\delta} - \Delta a^d}{M^d} \right) \tag{67}$$

$$= 1 - \delta \tag{68}$$

This finishes the proof of Theorem 1. $\qquad\square$

### A.7 PROOF OF THEOREM 2

In this section, we consider the performance of our policy as the sample size $N$ grows.

Let $d^{\mathcal{D}}$ denote the data distribution from which $\mathcal{D}$ is obtained. Thus $\mathcal{D}$ can be viewed as $N$ *i.i.d.* samples from $d^{\mathcal{D}}$. In this section, we use the notation with subscript $\mathcal{D}_N$ to denote $\mathcal{D}$ to address the number of data and avoid ambiguity.

Recall that $\pi_\theta$ minimizes the empirical objective $\frac{1}{N} \sum_{i=1}^{N} D_{\mathrm{KL}}(d^*(s_i)\pi_\theta(\cdot|s_i)\|d^*(s_i, \cdot))$ as an estimation in Eq.(14).

We make following assumptions:

**Assumption 3.** *Denote the space of parameter $\theta$ in the policy extraction step by $\Theta$. Let $g_\theta(s) := D_{\mathrm{KL}}(\pi_\theta(\cdot|s)\|\pi^*(\cdot|s))$, where $d^*(s)$ denotes the state marginal of $d^*$. Then, the function class $\mathcal{F} = \{g_\theta(\cdot) : \mathcal{S} \to \mathbb{R}|\theta \in \Theta\}$ is $d^{\mathcal{D}}$-Donsker. And $\mathrm{Var}_{s \sim d^{\mathcal{D}}(s)}(g_\theta(s)) < \infty$ for all $\theta \in \Theta$.*

Assumption 3 guarantees the consistency of $\theta$ trained with dataset $\mathcal{D}_N$, which is a common assumption when considering training with finite samples Van der Vaart (2000); Geer (2000); Ma & Kosorok (2005); Cheng & Huang (2010). A sufficient condition for Assumption 3 is $\Theta$ being bounded, together with a Lipschitz-type condition on $\mathcal{F}$ Van der Vaart (2000).

**Assumption 4.** *Suppose the policy extracted from Eq.(14) is $\pi$, define the state marginal of $d^{\mathcal{D}}, d^\pi, d^*$ as $d^{\mathcal{D}}(s), d^\pi(s), d^*(s)$, then*

$$D_{\mathrm{TV}}(d^\pi(s)\|d^*(s)) \leq D_{\mathrm{TV}}(d^{\mathcal{D}}(s)\|d^*(s)) \tag{69}$$

The Assumption 4 holds in general because the performance of learned policy $\pi$ is empirically in between $\pi^{\mathcal{D}}$ and $\pi^*$, indicating that the stationary state distribution of learned policy $d^\pi$ is closer to the optimal state distribution than dataset distribution.

Then we introduce the following lemma based on Lemma 6 in Xu et al. (2020):

**Lemma 2.** *Suppose the maximum reward is $R_{max} = \max_{s,a} \|r(s,a)\|$, $V^\pi(\rho_0) := \mathbb{E}_{s_0 \sim \rho_0}[V^\pi(s_0)]$ denote the performance given a policy $\pi$, then with assumption 4,*

$$|V^\pi(\rho_0) - V^*(\rho_0)| \leq \frac{2R_{max}}{1 - \gamma} D_{\mathrm{TV}}(d^*(s)\|d^{\mathcal{D}}(s)) + \frac{2R_{max}}{1 - \gamma} \mathbb{E}_{d^{\mathcal{D}}(s)}[D_{\mathrm{TV}}(\pi(\cdot|s)\|\pi^*(\cdot|s))], \tag{70}$$

*where $d^\pi(s), d^{\mathcal{D}}(s)$ denote the state marginal of $d^\pi, d^{\mathcal{D}}$ and $d^{\mathcal{D}}\pi(s, a) := d^{\mathcal{D}}(s)\pi(a|s)$.*

*Proof.*

$$|V^\pi(\rho_0) - V^*(\rho_0)| = \frac{1}{1-\gamma}\left|\mathbb{E}_{(s,a)\sim d^\pi}[r(s,a)] - \mathbb{E}_{(s,a)\sim d^*}[r(s,a)]\right| \tag{71}$$

$$\leq \frac{R_{\max}}{1-\gamma}\sum_{s,a}|d^\pi(s,a) - d^*(s,a)| \tag{72}$$

$$= \frac{2R_{\max}}{1-\gamma}D_{\mathrm{TV}}(d^\pi\|d^*) \tag{73}$$

$$\leq \frac{2R_{\max}}{1-\gamma}\left(D_{\mathrm{TV}}(d^\pi\|d^*(s)\cdot\pi) + D_{\mathrm{TV}}(d^*(s)\cdot\pi\|d^*)\right) \tag{74}$$

$$= \frac{2R_{\max}}{1-\gamma}D_{\mathrm{TV}}(d^\pi(s)\|d^*(s)) + \frac{2R_{\max}}{1-\gamma}\mathbb{E}_{d^*(s)}[D_{\mathrm{TV}}(\pi(\cdot|s)\|\pi^*(\cdot|s))] \tag{75}$$

$$\leq \frac{2R_{\max}}{1-\gamma}D_{\mathrm{TV}}(d^{\mathcal{D}}(s)\|d^*(s)) + \frac{2R_{\max}}{1-\gamma}\mathbb{E}_{d^*(s)}[D_{\mathrm{TV}}(\pi(\cdot|s)\|\pi^*(\cdot|s))] \tag{76}$$

The Eq.(74) follows the triangle inequality of TV distance. $\square$

Now we give the complete statement and proof of Theorem 2.

**Theorem 3.** *Suppose the maximum reward is $R_{max} = \max_{s,a}\|r(s,a)\|$, let $V^\pi(\rho_0) := \mathbb{E}_{s_0\sim\rho_0}[V^\pi(s_0)]$ denote the performance given a policy $\pi$. For policy $\pi_\theta$ optimized by Eq.(14) and $N$ transition data from $d^{\mathcal{D}}$, if $\pi_\theta$ is a universal approximator, under Assumption 3 and 4, we have*

$$V^*(\rho_0) - V^{\pi_\theta}(\rho_0) \leq \frac{2R_{max}}{1-\gamma}D_{\mathrm{TV}}(d^{\mathcal{D}}(s)\|d^*(s)) + e_N$$

*and $e_N$ converges in probability to zero at the rate $N^{-\frac{1}{4+h}}, \forall h > 0$, i.e., $N^{\frac{1}{4+h}}e_N \xrightarrow{N\to\infty} 0$ in probability.*

*Proof.* By Lemma 2, it remains to establish the vanishing rate of $e_N := \frac{2R_{\max}}{1-\gamma}\mathbb{E}_{d^{\mathcal{D}}}[D_{\mathrm{TV}}(\pi\|\pi^*)]$. By Pinsker's inequality and Jensen's inequality,

$$\mathbb{E}_{s\sim d^{\mathcal{D}}}[D_{\mathrm{TV}}(\pi(\cdot|s)\|\pi^*(\cdot|s))] \leq \mathbb{E}_{s\sim d^{\mathcal{D}}}[\sqrt{2D_{\mathrm{KL}}(\pi(\cdot|s)\|\pi^*(\cdot|s))}] \tag{77}$$

$$\leq \sqrt{2\mathbb{E}_{s\sim d^{\mathcal{D}}}[D_{\mathrm{KL}}(\pi(\cdot|s)\|\pi^*(\cdot|s))]} \tag{78}$$

Recall that the $\pi_\theta$ minimizes an empirical expectation

$$\min_\theta \frac{1}{N}\sum_{i=1}^N D_{\mathrm{KL}}(d^*(s_i)\pi_\theta(\cdot|s_i)\|d^*(s_i,\cdot)) = \frac{1}{N}\sum_{i=1}^N D_{\mathrm{KL}}(\pi_\theta(\cdot|s_i)\|\pi^*(\cdot|s_i)). \tag{79}$$

i.e., the objective is equivalent to minimizing the KL divergence over policy distribution. When $\pi_\theta$ is a universal approximator, it exactly minimizes the objective to 0.

Use notation $g_\theta(s) = D_{\mathrm{KL}}(\pi(\cdot|s)\|\pi^*(\cdot|s))$ as defined in Assumption 3. By Assumption 3, $\sqrt{N}(\mathbb{E}_{s\sim d^{\mathcal{D}}}[g_\theta(s)] - \frac{1}{N}\sum_{i=1}^N g_\theta(s_i))$ converges in distribution to a normal distribution with mean 0 and variance $\mathrm{Var}_{s\sim d^{\mathcal{D}}}(g_\theta(s)) < \infty$. (see e.g., Van der Vaart (2000))

As a result, for any $h > 0$,

$$N^{\frac{1}{2+h}}\left(\mathbb{E}_{s\sim d^{\mathcal{D}}}[g_\theta(s)] - \frac{1}{N}\sum_{i=1}^N g_\theta(s_i)\right) \xrightarrow{N\to\infty} 0, \quad \text{in probability} \tag{80}$$

Therefore,

$$\mathbb{E}_{s\sim d^{\mathcal{D}}}[D_{\mathrm{TV}}(\pi_\theta(\cdot|s)\|\pi^*(\cdot|s))] \leq \sqrt{2\mathbb{E}_{s\sim d^{\mathcal{D}}}[D_{\mathrm{KL}}(\pi_\theta(\cdot|s)\|\pi^*(\cdot|s))]} \tag{81}$$

$$= \sqrt{2\mathbb{E}_{s\sim d^{\mathcal{D}}}[g_\theta(s)]} \tag{82}$$

$$= \sqrt{2\left(\mathbb{E}_{s\sim d^{\mathcal{D}}}[g_\theta(s)] - \frac{1}{N}\sum_{i=1}^N g_\theta(s)\right)} \tag{83}$$

Combine with equation 80, for any $h > 0$, we have

$$N^{\frac{1}{4+h}} e_N = N^{\frac{1}{4+h}} \frac{2R_{\max}}{1-\gamma} \mathbb{E}_{s \sim d^{\mathcal{D}}}[D_{\mathrm{TV}}(\pi(\cdot|s) \| \pi^*(\cdot|s))] \tag{84}$$

$$\leq \frac{2R_{\max}}{1-\gamma} \sqrt{2N^{\frac{1}{2+h/2}} \left( \mathbb{E}_{s \sim d^{\mathcal{D}}}[g_\theta(s)] - \frac{1}{N} \sum_{i=1}^{N} g_\theta(s) \right)} \xrightarrow{N \to \infty} 0, \text{ in probability} \tag{85}$$

This finishes the proof. $\qquad\square$

# B    EXPERIMENT DETAILS

## B.1    TASKS AND BASELINES

**Tasks.** We adopt "-v1" tasks for Maze2D and Adroit domains and "-v2" tasks for MuJoCo domain. We do not adopt antmaze tasks because the initial distributions are significantly different between training and test settings: the starting point can be anywhere in the offline dataset but is always at the same corner when tested. CDE learns the value functions by optimizing an objective (eq.(6)) w.r.t. initial distribution and $d^{\mathcal{D}}$, which is not compatible with antmaze tasks. This inconsistency also partially explain why trajectory optimization methods (e.g., decision transformer) fail but the Bellman bootstrapping methods are less affected.

We make the rewards sparse in MuJoCo tasks as stated in main text, and return thresholds (i.e., the 75-percentile of the trajectory returns in dataset) are listed in table 3. During evaluation, a trajecotory is viewed as a successful one if its reward return exceeds the threshold.

Table 3: The return thresholds for sparse-MuJoCo tasks.

| Task | Return threshold |
|---|---|
| halfcheetah-medium | 4909.1 |
| walker2d-medium | 3697.8 |
| hopper-medium | 1621.5 |
| halfcheetah-medium-expert | 10703.4 |
| walker2d-medium-expert | 4924.8 |
| hopper-medium-expert | 3561.9 |

**Details of baselines.** We adopt the results of baselines if reported in the original paper. We rerun the baselines for these tasks using their official codes (IQL, TD3+BC, OptiDICE) or the d3rlpy library (Seno & Imai, 2022) (BCQ, CQL), because 1) d3rlpy keeps the same hyperparameters as the original papers, and 2) the performance of d3rlpy is better and more stable than the original implementation for BCQ, CQL (e.g., the performances of BCQ on Maze2d and Adroit).

## B.2    FULL ALGORITHM AND PRACTICAL DETAILS OF CDE

In this section, we present the full algorithm and implementation details. Without otherwise statements, the policies or critics defaults to be parameterized by neural networks (NN).

### B.2.1    VALUE FUNCTIONS SEPARATION

In CDE, we learn both the V-value function and the advantage function. The former can incorporate the stochasticity of action distribution to reduce the instability, and the latter is to generalize the optimal importance ratios to OOD regions since the reward and transition probability functions for unseen transition $(s, a, r, s')$ are absent in offline datasets. Meanwhile, we take two steps to train V-value and advantage functions instead of optimizing the objective function in Eq.(6). Note that the objective can be separated into in-distribution and OOD parts:

$$\zeta \mathbb{E}_{d^{\mathcal{D}}} \left[ w(s,a)A(s,a) - \alpha f(w(s,a)) \right] + (1-\gamma)\mathbb{E}_{\rho_0}[v(s_0)]$$
$$+ (1-\zeta)\mathbb{E}_\mu \left[ w(s,a)(A(s,a) - \lambda(s,a)) - \alpha f(w(s,a)) + \tilde{\epsilon}\lambda(s,a) \right], \tag{86}$$
$$= \zeta(\mathbb{E}_{d^{\mathcal{D}}} \left[ w(s,a)A(s,a) - \alpha f(w(s,a)) \right] + ((1-\gamma)\mathbb{E}_{\rho_0}[v(s_0)]))$$
$$+ (1-\zeta)(\mathbb{E}_\mu \left[ w(s,a)(A(s,a) - \lambda(s,a)) - \alpha f(w(s,a)) + \tilde{\epsilon}\lambda(s,a) \right] + (1-\gamma)\mathbb{E}_{\rho_0}[v(s_0)]) \tag{87}$$

where the in-distribution part corresponds to the learning objective of the V-value function in Eq.(10), which is also the dual form of following constrained optimization:

$$\max_{d^\pi \geq 0} \mathbb{E}_{d^\pi}[r(s,a)] - \alpha D_f(d^\pi \| d^{\mathcal{D}}) \tag{88}$$

$$s.t. \sum_a d^\pi(s,a) = (1-\gamma)\rho_0 + \mathcal{T}_* d^\pi(s), \forall s, a \in \text{supp}(d^{\mathcal{D}}). \tag{89}$$

The main difference of it from the previous one in Eq.(1) is that it constrains the state-action in the support of offline datasets. Therefore, the objective for V-value function learning in Eq.(10) is still a convex optimization problem.

### B.2.2 POLICY EXTRACTION

In policy extraction, we need to estimate the KL divergence between learned policy and mixed behavior policy $\hat{\pi}^{\mathcal{D}}$ in eq.(14). As we only train the behavior policy $\pi^{\mathcal{D}}$ from offline dataset, we apply Jensen inequality to get an upper bound of the objective:

$$D_{\mathrm{KL}}[\pi_\theta(\cdot|s)\|\hat{\pi}^{\mathcal{D}}(\cdot|s)] = \mathbb{E}_{a\sim\pi_\theta}[\log\pi(a|s) - \log(\zeta\pi^{\mathcal{D}}(\cdot|s) + (1-\zeta)\pi^\mu(a|s))] \tag{90}$$

$$\leq \mathbb{E}_{a\sim\pi_\theta}[\log\pi(a|s) - \zeta\log\pi^{\mathcal{D}}(\cdot|s) - (1-\zeta)\log\pi^\mu(a|s)] \tag{91}$$

which is adopted in policy extraction step in practical implementation.

Meanwhile, we simply set the optimal state distribution $d^*(s)$ as a uniform distribution on the states in the successful trajectories (i.e., the trajectories with returns larger than 0).

### B.2.3 OOD ACTION SPACE

Remember we define the OOD action region $\mathcal{A}_{\mathrm{OOD}}$ based on the $\Delta a$ in Sec. 3.2. Here $\Delta a$ defines the radius of the in-distribution action region and $\mathcal{A}_{\mathrm{OOD}}$ can cover the action space without overlapping with in-distribution action. As we cannot precisely tell the in-distribution region given an offline dataset, which only provides one action for each state especially in continuous tasks. Therefore, we estimate it by behavioral cloning: we employ NN to approximate the behavior policy $\pi^{\mathcal{D}}$ that outputs a Gaussian distribution $\mathcal{N}(\mu^{\mathcal{D}}(s), \sigma^{\mathcal{D}2}(s))$ for each state; then the standard deviation $\sigma^{\mathcal{D}}(s)$ can be a measurement for the breadth of in-distribution region on. In practice, we adopt $\Delta a = \sigma^{\mathcal{D}}(s)$ when computing $\mathcal{A}_{\mathrm{OOD}}(s)$. We further give a parameter study on $\Delta a$ in appendix B.4.3.

### B.2.4 HYPERPARAMETERS

Before training NN, we standardize the observation and reward and scale the reward by multiplying $0.1$. To extract the policy after the optimization over value functions converges, we set the warm-up training step following Lee et al. (2021) and the policy $\pi_\theta$ will not start until warm-up training ends. As shown in table 4, we set the same $f$-divergence coefficient $\alpha$ for each domain because the divergence between the optimal and behavior policies varies in different domains. Note this is still *significantly different from previous DICE paper (Lee et al., 2021) that finetunes and assigns different hyperparameters for every single task*. The other shared hyperparameters are summaries in table 5. More details can be found in codes provided in supplementary materials.

Table 5: The shared hyperparameters.

Table 4: The $f$-divergence coefficient $\alpha$.

| Domain | $\alpha$ |
|---|---|
| Maze2D | 0.001 |
| Adroit | 0.01 |
| MuJoCo | 0.1 |

| Hyperparameters | values |
|---|---|
| hidden layers of policy $\pi_\theta$ | [256,256] |
| hidden layers of $\pi^{\mathcal{D}}$ | [256,256] |
| number of mixtures of $\pi^{\mathcal{D}}$ | 3 |
| hidden layers of V-value $v_\varphi$ | [256,256] |
| hidden layers of advantage $A_\phi$ | [256,256] |
| activation function of networks | ReLU |
| NN optimizer | Adam |
| NN learning rate | 3e-4 |
| discount factor $\gamma$ | 0.99 |
| batch size | 512 |
| mixture coefficient $\zeta$ | 0.9 |
| max OOD IS ratio $\tilde{\epsilon}$ | 0.3 |
| number of OOD action samples | 5 |

### B.2.5 FULL ALGORITHM

In this section, we present the full algorithm in Algorithm. 2.

---

**Algorithm 2** Full Algorithm of Conservative Density Estimation

---

Initialize value functions $v_\varphi$, $\tilde{A}_\phi$, behavior policy $\pi^{\mathcal{D}}$, policy $\pi_\theta$. All the advantage functions will be subtracted by $\eta$ during computation as proved in Appendix A.2.

1: **for** training iteration $i$ **do**
2:     ▷ *policy evaluation and improvement*
3:     Sample batch $\{(s_i, a_i, r_i, s'_i)\}$ from $\mathcal{D}$ and $n$ OOD actions $\{a^{(1)}, \dots, a^{(n)}\}$ for each $s$;
4:     Update V-value $v_\varphi$ by Eq.(10);
5:     Compute in-distribution advantage function $A_\varphi(s, a)$ via $v_\phi$;
6:     Update regularized advantage $\tilde{A}_\phi$ by Eq.(11);
7:     Update distribution normalizer $\eta$ by gradient descent: $\eta \leftarrow \eta - \alpha_\eta (1 - \mathbb{E}[w^*(s, a)])$.
8:     Update $\pi^{\mathcal{D}}$ by behavioral cloning.
9:     ▷ *policy extraction*
10:     **if** $i \geq$ warm-up steps **then**
11:         Update policy $\pi_\theta$ by Eq.(14) with entropy regularization.
12:     **end if**
13: **end for**

---

### B.3 MORE EXPERIMENT RESULTS

#### B.3.1 TRAINING CURVES

The training curves of full dataset experiments are shown in fig. 4. The training steps start from a non-zero number because of the warm-up step, i.e., we learn the policy after the value function almost converges. The warm-up step is 20,000 for the maze2d environment and 40,000 for other environments.

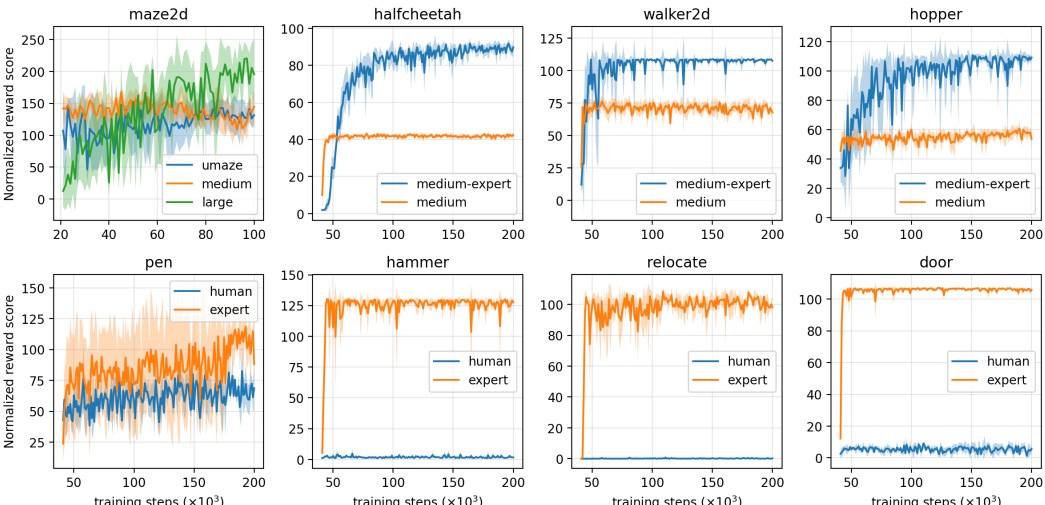

Figure 4: The training curves of CDE. The shadow region indicates the standard deviation of mean values across different seeds. Here we report the normalized reward scores for MuJoCo tasks measured by dense rewards instead of success rate, which has been reported in previous tables.

We can observe that our method converges extremely fast and is very stable during training. This is because CDE employs convex optimization to solve the value function and extracts the optimal policy in a manner of supervised learning. On the contrary, the previous methods (e.g., Q-learning-based methods (Kumar et al., 2020; Kostrikov et al., 2021b)) are prone to over-fitting in training due to the interleaved optimization of value and policy (Brandfonbrener et al., 2021), which may lead to large compounded errors and performance decrease especially with long training steps.

### B.3.2 THE EXPERIMENT RESULTS ON MORE TASKS

We present more comparisons on MuJoCo "-medium-replay" tasks and Adroit "-cloned" tasks in table 6, 7 respectively.

Table 6: Success rate (%) of CDE against other baselines on sparse-MuJoCo "-medium-replay" tasks.

|  | BCQ | CQL | IQL | TD3+BC | CDE (Ours) |
|---|---|---|---|---|---|
| halfcheetah-medium-replay | 91.0±2.2 | 48.0±14.9 | 70.7±3.7 | 61.0±16.3 | **95.2**±3.5 |
| walker2d-medium-replay | 85.7±8.4 | 78.3±11.2 | 4.0±4.9 | 84.4±7.3 | **90.4**±5.0 |
| hopper-medium-replay | 91.4±5.2 | 87.0±5.1 | 0.0±0.0 | 89.0±4.1 | **93.3**±4.6 |

Table 7: Normalized scores of CDE against other baselines on Adroit "-cloned" tasks.

|  | BCQ | CQL | IQL | CDE (Ours) |
|---|---|---|---|---|
| pen-cloned | 44.0 | 39.2 | 37.3 | **56.9**±13.2 |
| hammer-cloned | 0.4 | 2.1 | 2.1 | **3.3**±1.7 |
| door-cloned | 0.0 | 0.4 | **1.6** | 0.3±0.1 |
| relocate-cloned | -0.3 | -0.1 | -0.2 | **0.4**±0.3 |

By the results listed above, we can observe that CDE exceeds most baselines on sparse MuJoCo "-medium-replay" and Adroit "-cloned" tasks.

### B.3.3 COMPARISON ON SPARSE MEDIUM-EXPERT MUJOCO TASKS IN SCARCE DATA SETTING

The experiment results on MuJoCo medium-expert tasks are shown in fig. 5. CDE obtains similar performances to OptiDICE and IQL on halfcheetah and hopper tasks and exceeds all others on walker2d task. Meanwhile, CDE has higher success rate with a small proportion of data.

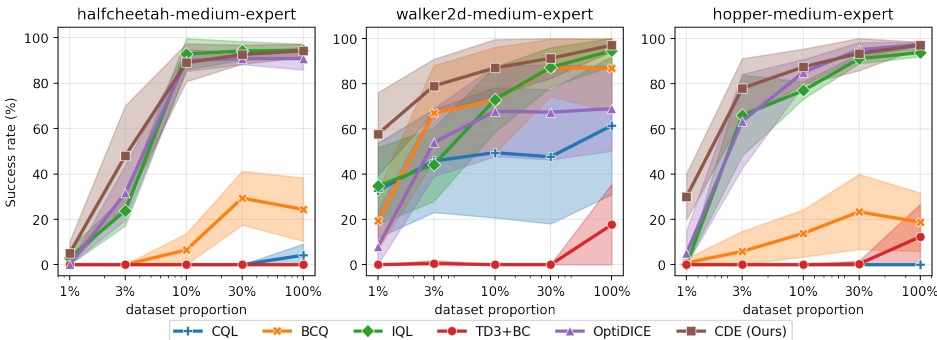

Figure 5: The results on sub-datasets with different dataset sizes for MuJoCo medium-expert tasks.

### B.3.4 THE EXPERIMENT RESULTS ON ORIGINAL D4RL MUJOCO TASKS

We provide the performances of CDE and comparison with baselines on dense-reward MuJoCo tasks in table 8.

We can observe that while CDE may not outperform the baselines in dense reward settings, it achieves a significantly higher success rate when reward is sparse, which indicates that the baseline performances are predominantly reliant on dense rewards and highlights the effectiveness of CDE for sparse-reward offline data.

### B.3.5 PERFORMANCES ON SPARSE-REWARD GOAL-CONDITIONED TASKS

To further illustrate the effectiveness of CDE on sparse-reward tasks, we test it on goal-reaching tasks from Yang et al. (2022), where the agent receives a "+1" reward only if it reaches a desired goal and a 0 reward otherwise. Follow Ma et al. (2022), we choose 5 relatively hard tasks and the offline

Table 8: Normalized scores on original dense-reward MuJoCo tasks.

| | BCQ | CQL | IQL | TD3+BC | CDE |
|---|---|---|---|---|---|
| halfcheetah-medium | 40.7 | 49.1 | 47.4 | 48.3 | 43.3±2.9 |
| walker2d-medium | 54.5 | 82.9 | 78.3 | 83.7 | 73.8±4.4 |
| hopper-medium | 53.1 | 64.6 | 66.3 | 59.3 | 51.2±3.7 |
| halfcheetah-medium-expert | 64.7 | 85.8 | 86.7 | 90.7 | 75.6±7.2 |
| walker2d-medium-expert | 110.9 | 109.5 | 109.6 | 110.1 | 107.7±10.4 |
| hopper-medium-expert | 57.5 | 102.0 | 91.5 | 98.0 | 108.6±4.8 |

datasets consist of expert and random data. We then compare our method with GCSL (Ghosh et al., 2019), WGCSL (Yang et al., 2022) and GoFAR (Ma et al., 2022), where the former two methods are goal-conditioned imitation learning and the latter one is a goal-conditioned version of DICE method.

Table 9 shows the average reward returns comparison. The performances of baselines are adopted from Ma et al. (2022). Note that all compared baselines use goal relabeling during training while our method ignores the goal and only uses the states to train policy. We can find that CDE obtains the highest rewards on 3 tasks and comparable rewards on FetchPush. Meanwhile, CDE does not use goal state during training and may fail to stitch transitions from different trajectories, which can be a part of reason for failure on FetchSlide. As this paper focuses on the standard offline RL setting, it can be a future direction to extend existing method to goal-conditioned setting.

Table 9: The performances comparison on goal-conditioned tasks.

| Task | GCSL | WGCSL | GoFAR | CDE |
|---|---|---|---|---|
| FetchReach | 20.9±2.8 | 21.9±2.1 | 28.2±0.6 | **29.1±1.7** |
| FetchPick | 8.9±3.1 | 9.8±2.6 | 19.7±2.6 | **27.7±1.4** |
| FetchPush | 13.4±3.0 | 14.7±2.7 | **18.2±3.0** | 16.6±2.0 |
| FetchSlide | 1.8±1.3 | **2.7±1.6** | 2.5±1.4 | 1.1±1.0 |
| HandReach | 1.4±2.2 | 6.0±4.8 | 11.5±5.3 | **17.0±2.9** |

### B.3.6 COMPARISON ON COMPUTATION TIME

One drawback of our method compared to baselines is that CDE may require more computation time in training. Therefore, we compare baselines and our method on halfcheetah-medium-expert-v2 task. We use the server with AMD EPYC 7542 32-Core CPU and A5000 GPU. To compare different methods fairly, we report the time of 200,000 steps with batch size = 512 and defaulted hyper-parameters for baselines. The final results are shown in table 10.

Table 10: The computation time comparison.

| Method | BCQ | CQL | IQL | TD3+BC | CDE |
|---|---|---|---|---|---|
| Computation time (min) | ∼200 | ∼150 | ∼40 | ∼30 | ∼90 |

### B.4 MORE PARAMETER STUDIES AND ABLATION STUDIES

In this section, we present more parameter studies w.r.t the hyperparameters in our method. We mainly choose Maze2d tasks as our testbench as they are more sensitive to the hyperparameters.

### B.4.1 PARAMETER STUDY ON MAX OOD IS RATIO

Although we visualize the stationary state distribution of policies with different $\tilde{\epsilon}$ in maze2d-large task, we provide performances on all maze2d tasks in table 11 as a supplement. As $\tilde{\epsilon}$ decreases, constraint on unseen region gets stronger. There is a performance gain with proper conservatism but it also ruins the final performance if the constraint is too conservative.

Table 11: The performances on maze2d tasks with different $\tilde{\epsilon}$.

| $\tilde{\epsilon}$ | 3.0 | 0.3 | 0.03 | 0.003 |
|---|---|---|---|---|
| maze2d-umaze | 121.7±12.5 | 134.1±10.4 | 132.7±11.4 | 114.5±17.3 |
| maze2d-medium | 154.2±14.9 | 146.1±13.1 | 135.2±12.3 | 89.4±29.5 |
| maze2d-large | 189.7±19.7 | 210.0±13.5 | 174.5±24.7 | 51.6±37.9 |

### B.4.2 PARAMETER STUDY ON NUMBER OF OOD ACTION SAMPLES

We use sampling on OOD actions in eq.(11) to approximate the out-of-distribution constraint. To study how the action sample number $N$ influences the final performance, we test our method with different $N$ on maze2d tasks and show the results in table 12. Theoretically, the approximation error on OOD constraint will decrease as we increase the $N$. In practice, we find there is no significant performance improvement when $N \geq 3$. Therefore, we simply set $N = 5$ for all tasks in practice.

Table 12: The performances on maze2d tasks with different number of OOD action samples $N$.

| N | 1 | 3 | 5 (adopted) | 10 |
|---|---|---|---|---|
| maze2d-umaze | 115.2±6.9 | 131.7±3.5 | 134.1±10.4 | 141.1±10.1 |
| maze2d-medium | 132.6±11.9 | 142.9±13.8 | 146.1±13.1 | 146.8±14.8 |
| maze2d-large | 179.4±21.5 | 202.0±24.7 | 210.0±13.5 | 209.7±14.8 |

### B.4.3 PARAMETER STUDY ON IN-DISTRIBUTION WIDTH

We use the standard deviation of the behavior policy output $\sigma^{\mathcal{D}}(s)$ as a measurement of the width of in-distribution region. However, there may still be some concerns on the overlap between OOD action space and in-distribution region. Therefore, we conduct a parameter study on $\Delta a$.

Table 13: The performances on maze2d tasks with different $\Delta a$.

| $\Delta a / \sigma^{\mathcal{D}}(s)$ | 0.3 | 0.7 | 1.0 (adopted) |
|---|---|---|---|
| maze2d-umaze | 131.3±16.4 | 137.7±9.3 | 134.1±10.4 |
| maze2d-medium | 149.1±12.7 | 151.2±11.0 | 146.1±13.1 |
| maze2d-large | 211.5±14.3 | 204.0±17.3 | 210.0±13.5 |

As listed in table 13, there is no significant performance drop when decreasing $\Delta a / \sigma^{\mathcal{D}}(s)$ (i.e., the higher risk of overlap between OOD and in-distribution region). This is because the advantage learning step in eq.(11) involves both in-distribution and OOD regression. When an in-distribution action is wrongly recognized as OOD action, its value function will exist both in two terms. Therefore, the value function of in-distribution action will not be negatively affected while the OOD actions in $\mathcal{A}_{ood}$ can always be constrained.

From the above results, we can observe that the performance of our method is robust to the most hyperparameters except $\tilde{\epsilon}$, which controls the degree of conservativeness on unseen regions.

### B.4.4 ABLATION STUDY ON WARM-UP STAGE

We adopt warm-up where we only update the value function and stop policy updating. To investigate its influence on final performances, we present the results of ablation study on Maze2d tasks in table 14.

Table 14: The performances on maze2d tasks with/without warm-up stage.

|                | with warm-up | w.o. warm-up |
| --- | --- | --- |
| maze2d-umaze   | 134.1±10.4   | 136.2±13.2   |
| maze2d-medium  | 146.1±13.1   | 141.7±17.3   |
| maze2d-large   | 210.0±13.5   | 205.4±15.4   |

We can find that the warmup stage does not influence the final performances significantly.

