# OpenReview forum: "Learning from Sparse Offline Datasets via Conservative Density Estimation"
_ICLR.cc/2024/Conference — ICLR 2024 poster_

### Official Review · Reviewer_6pFf · 2023-10-31

**Soundness:** 3 good
**Presentation:** 3 good
**Contribution:** 4 excellent
**Rating:** 8
**Confidence:** 3

**Summary:**

The paper proposes an Offline RL algorithm called Conservative Density Estimation (CDE) incorporating pessimism within the stationary distribution space. Extensive empirical study shows that CDE outperforms existing baselines in both the sparse-reward settings and the scarce-data settings.

**Strengths:**

The authors present a novel method of mixing pessimistic-based methods with DICE-style methods. Unlike other pessimistic methods that balance pessimism using a hyperparameter called a penalty coefficient, CDE finds the optimal balance based on theoretical grounds. The paper also introduces a new way to convert dense-reward MuJoCo tasks into a sparse-reward setting based on the trajectory return. Together with the scarce-data setting experiments, the two additional benchmarks can help test the robustness of offline RL algorithms. Finally, the arguments provided by the authors are mathematically sound.

**Weaknesses:**

1. §3.2.1 states that the strong duality holds due to Slater's condition. The fact that the optimization problem of our interest satisfies Slater's condition does not seem trivial.

2. The RHS of (7) may not exist. For example, $\tilde{A}(s, a)$ might be negative.

3. According to §B.2.2, the authors set $d^*(s)$ as a uniform distribution on the state in the successful trajectories. It would have been better if this explanation were included in the main paper. I am doubtful that this distribution will be close to $\tilde{w}^*(s, a)\hat{d}^{\mathcal{D}}(s, a)$, though.

### Minor comments

1. (18), (19) ⇒ the summation should also be over $s$ and $s'$, respectively

2. (29) ⇒ $d^{\mathcal{D}}$ in the second expectation → $\mu$

**Questions:**

1. How are the OOD actions sampled? Do you just sample a random action and reject it if it is in distribution?

2. Does Slater's condition also hold for optimization problems dealing with infinite-dimensional vectors (i.e., functions)?

---

> ### Author Response · Authors · 2023-11-16
> **Response to Reviewer 6pFf**
>
> We thank the reviewer for careful review and valuable suggestions.
>
> - **W1. Why does the Slater's condition of optimization problem in eq.(2)~(4) hold.**
>
>   Thanks for pointing out this question. Here the Slater's condition is that there exists a strictly feasible $d(s,a)$ s.t. the equality constraint in eq.(3) holds while constraint in eq.(4) is satisfied with strict inequality. One strictly feasible solution is empirical dataset distribution $d^D(s,a)$: (1) Bellman flow constraint $\sum_a d^D(s,a)=(1-\gamma)\rho_0(s) + \mathcal{T}_* d^D(s)$ holds because the dataset is sampled from real environment; (2) $d^D(s,a)=0<\epsilon \mu(s,a),\forall s,a\in\text{supp}(\mu)$ by definition of $\mu$. Therefore, the Slater's condition holds.
>
>   We have added it in appendix A.1 in revision to make our statement more clear.
>
> - **W2. RHS of eq.(7) may not exist when $\tilde{A}(s,a)<0$**
>
>   Actually, it still holds when $\tilde{A}(s,a)<0$ because we assume $(f')^{-1}(x)>0, \forall x\in \mathbb{R}$ in Assumption 1.
>   In practice, we adopt soft-chi function as $f$ as mentioned in **training details** in page 7, where
>   $$
>   f_{\text{soft}-\chi^2}(x) =
>   \begin{cases}
>   x\log x -x +1, 0<x<1\\\\
>   (x-1)^2/2,x\geq 1
>   \end{cases}
>   $$
>   Therefore,
>   $$
>   (f')^{-1}(x) =
>   \begin{cases}
>   e^{x},x<0\\\\
>   x+1,x\geq 0
>   \end{cases}
>   $$
>   This satisfies the above assumption and thus we still can compute RHS of eq.(7) when $\tilde{A}(s,a)<0$.
>
> - **W3. Set $d^{*}(s)$ as a uniform distribution on the state in the successful trajectories. Why is it close to $\tilde{w}(s,a)\hat{d}^D(s,a)$.**
>
>   $d^*(s)$ denotes the state distribution of optimal policy $\pi^*$, and $\pi^*$ should always succeed in any trajectory. So we believe it is reasonable to approximate $d^*(s)$ as the empirical distribution of the given successful trajectories.
>
>   The empirical distribution can be close to the state margin of $d^*(s,a)$ (i.e., $\sum_a \tilde{w}(s,a)\hat{d}^D(s,a)$): the state-action pair $(s,a)$ in failed trajectories should have a smaller importance ratio $\tilde{w}(s,a)$ than $(s,a)$ in successful trajectories; therefore, the density of $s$ from successful trajectories should also be much larger than $s$ from failure ones. Meanwhile, since it is infeasible to do summation of $\sum_a \tilde{w}(s,a)\hat{d}^D(s,a)$ over $a$ in continuous space to directly compute the state margin, we use the approximation for easy computation.
>
>
> - **Q1. How are the OOD actions sampled? Do you just sample a random action and reject it if it is in distribution?**
>
>   Yes, we uniformly sample an action and reject it if it is in distribution.
>
> - **Q2. Does Slater's condition also hold for optimization problems dealing with infinite-dimensional vectors (i.e., functions)?**
>
>   Yes. In fact, the stationary distribution $d:\mathcal{S}\times\mathcal{A}\rightarrow [0,1]$ is indeed a function if we consider continuous state-action space, which is the case of our experiment. Meanwhile, we can observe that the empirical dataset distribution $d^D(\cdot, \cdot)$ is still strictly feasible for optimization in eq.(2)~(4). Therefore, the Slater's condition still holds.
>
> - **Minor issues.**
>
>   Thanks for your careful check, we have corrected them in revision.
>
> Thanks again for your feedback. We hope our response addresses your main concerns.

---

> > ### Comment · Reviewer_6pFf · 2023-11-22
> >
> > Thank you for your detailed response.
> >
> > **W1** Under stochastic transition dynamics, I believe the Bellman flow constraint does not hold for $d^{\mathcal{D}}$ because the LHS is a discrete distribution while the RHS is a continuous one.
> >
> > **W2** I thought the assumption was $f'(x)\ge 0$ for all $x$. Sorry for the confusion.

---

> > > ### Author Response · Authors · 2023-11-22
> > > **Response to further question from Reviewer 6pFf**
> > >
> > > Thanks for raising this further question. We admit there will be mismatch if we view $d^D$ as a collection of point masses on sampled $(s,a)$ points (in practice, we assume $d^D>0$ also holds for those $(s,a)$ that are close to the sample state-action pairs, i.e., $(s,a)$ in in-distribution region). However, to make the Slater's condition hold, we still have a strictly feasible solution to the constrained optimization problem. Let $d^{\pi_D}$ denote the stationary state-action distribution of behavior policy $\pi_D$ of offline dataset, then $\sum_a d^{\pi_D}(s,a)$ is a continuous distribution over state and it can satisfy the Bellman flow constraint.
> > >
> > > We sincerely appreciate your careful review and hope this address your concerns.

---

### Official Review · Reviewer_MKs2 · 2023-11-01

**Soundness:** 2 fair
**Presentation:** 2 fair
**Contribution:** 2 fair
**Rating:** 5
**Confidence:** 4

**Summary:**

The paper proposes a density estimation-based offline RL algorithm that aims to incorporate the strengths of value penalization-based offline RL methods and DICE-based offline RL methods. The proposed method outperforms the certain baselines in considered sparse reward D4RL tasks.

**Strengths:**

- The proposed method outperforms the considered baselines significantly in maze2d tasks.
- The motivation which is to achieve the best of both worlds (between the value penalization methods and the DICE-based methods) is solid.

**Weaknesses:**

- Some clarifications are required in the introduction section. 1) "pessimism-based techniques may be prone to over-pessimism, especially in high-dimensional state-action spaces" => why is over-pessimism a bigger problem for high-dimensional state-action spaces? 2) "regularization methods often struggle with the tuning of the regularization coefficient" => Why does the author think this is true? The cited paper seems to be about DICE and not about regularization methods. Additionally, IQL (which is a baseline considered in the paper) uses the same hyperparameters for experiments in the same domain, just like the proposed method.
- The considered tasks are not thorough. First, there are no experiment results on D4RL Gym *-medium-replay datasets, which contain trajectories that vary significantly in performance. These datasets can test if the proposed method can handle datasets with mixed quality. Second, on Adroit, I am curious on why the authors did not include the *-cloned datasets while including *-human datasets. Third, some important tasks with sparse reward are missing: Antmaze and Kitchen. These tasks are much harder than simple maze2d and at the same time have sparse rewards.
- Some important baselines are missing. To name a few, DecisionTransformer [1] and Diffuser [2] also claim they have strength in the sparse reward settings.

[1] Chen et al., Decision Transformer: Reinforcement Learning via Sequence Modeling, NeurIPS 2021.

[2] Janner et al., Planning with Diffusion for Flexible Behavior Synthesis, ICML 2022.

**Questions:**

- On sparse-MuJoCo tasks, how does the proposed method and the baselines perform if you change the cut-off percentile (which is currently 75)?
- How does the proposed method perform compared to the baselines on dense reward settings?

---

> ### Author Response · Authors · 2023-11-16
> **Response to Reviewer MKs2 (1/2)**
>
> We thank the reviewer for valuable feedback.
>
> - **W1. Some clarifications are required in the introduction section.**
>
>   - **Why is over-pessimism a bigger problem for high-dimensional state-action spaces?**
>
>     Our statement is based on (1) the accurate data distribution estimation is more difficult when space is high-dimensional, and (2) it is relatively harder to overcome over-pessimism issue with large estimation error on data distribution [1][2]. But we also concede that the over-pessimism is not necessarily more severe when space is high-dimensional. We have modified our statements accordingly and thanks for pointing it out.
>
>   - **Why do regularization methods struggle with the tuning of the regularization coefficient?**
>
>     We do agree many offline RL methods can achieve good performances on different tasks with similar hyper-parameters such as IQL or TD3+BC. However, there are also some methods [3][4] requiring finetuning the regularization coefficient.
>
>     Although the cited paper is DICE-based, it adds a f-divergence regularization in the objective. This paper also finetunes the coefficient in different tasks.
>
>   Thanks for your careful check. We update statements in revision and make them more rigorous now.
>
>
> - **W2. The considered tasks are not thorough.**
>
>   Thanks for your questions. We add more experiment results as follows.
>   - Mujoco medium-replay tasks.
>     |                 | BCQ      | CQL       | IQL      | TD3+BC    | CDE (Ours)   |
>     | --------------- | -------- | --------- | -------- | --------- | ------------ |
>     | halfcheetah-m-r | 91.0±2.2 | 48.0±14.9 | 70.7±3.7 | 61.0±16.3 | **95.2±3.5** |
>     | walker2d-m-r    | 85.7±8.4 | 78.3±11.2 | 4.0±4.9  | 84.4±7.3  | **90.4±5.0** |
>     | hopper-m-r      | 91.4±5.2 | 87.0±5.1  | 0.0±0.0  | 89.0±4.1  | **93.3±4.6** |
>
>     CDE exceeds the compared baselines on medium-replay tasks.
>
>   - Adroit cloned tasks.
>     |                 | BCQ  | CQL  | IQL     | CDE (Ours)    |
>     | --------------- | ---- | ---- | ------- | ------------- |
>     | pen-cloned      | 44.0 | 39.2 | 37.3    | **56.9±13.2** |
>     | hammer-cloned   | 0.4  | 2.1  | 2.1     | **3.3±1.7**   |
>     | door-cloned     | 0.0  | 0.4  | **1.6** | 0.3±0.1       |
>     | relocate-cloned | -0.3 | -0.1 | -0.2    | **0.4±0.3**   |
>
>     We did not include them in our original manuscript because both the offline data (collected by a policy which is trained by behavioral cloning on human "tasks") and final performances are similar to "human" tasks. Our method still obtains the best or comparable performances on most tasks. We have added the new experiments in appendix B.3.2 in revision.
>
>   - Antmaze and kitchen tasks.
>
>     In antmaze and kitchen tasks, the initial distributions are significantly different between training and test settings, which violates the basic assumption of CDE. For the antmaze tasks, the starting point can be anywhere in the offline dataset but is always at the same corner when tested. The mismatch also happens on kitchen tasks. On the contrary, the optimization objective of CDE (eq.(6)) is given by initial distribution $\rho_0$ and $d^D$, requiring the consistency of initial distribution. As we mentioned in "Conclusion" section, one direction to overcome this inconsistency is to incorporate goal-conditioned setting, which we leave as a future work.
>
> - **W3. Some baselines are missing, e.g., DT and Diffuser.**
>
>   Thanks for your suggestion on baselines. We add comparison as follows (we report normalized score for maze2d and success rate for sparse mujoco).
>   |                           | Decision Transformer | Diffuser     | CDE (Ours)     |
>   | ------------------------- | -------------------- | ------------ | -------------- |
>    maze2d-umaze              | 21.1±25.4            | 113.9±3.1    | **134.1±10.4** |
>   | maze2d-medium             | 29.7±13.3            | 121.5±2.7    | **146.1±13.1** |
>   | maze2d-large              | 41.7±14.1            | 123.0±6.4    | **210.0±13.5** |
>   | halfcheetah-medium        | 39.7±1.9             | 66.7±10.2    | **82.0±8.6**   |
>   | walker2d-medium           | 10.3±2.1             | 46.3±16.7    | **53.0±11.7**  |
>   | hopper-medium             | 63.0±19.9            | **94.0±2.8** | 85.5±5.7       |
>   | halfcheetah-medium-expert | 0.0±0.0              | 0.0±0.0      | **95.2±2.9**   |
>   | walker2d-medium-expert    | 0.0±0.0              | 74.0±11.0    | **97.0±2.8**   |
>   | hopper-medium-expert      | 0.0±0.0              | 92.7±5.0     | **97.0±1.4**   |
>
>   Our method exceeds DT and diffuser on most tasks. Despite of its notable performances in dense reward setting, the performances of DT drop remarkably in sparse reward setting, indicating DT heavily relies on dense reward to train the transformer and supervise policy learning.

---

> ### Author Response · Authors · 2023-11-16
> **Response to Reviewer MKs2 (2/2)**
>
> - **Q1. The performances on sparse-mujoco with different cut-off percentiles.**
>
>   Since CDE exceeds the baselines by a large margin on "-medium-expert" tasks, we test the performances with 90 percentile and 50 percentile as cut-off thresholds on "-medium" level tasks. The results are as follows:
>
>   **The success rate of CDE and baselines with cut-off percentile=90**
>   |                    | BCQ       | CQL          | IQL      | TD3+BC    | CDE (Ours)    |
>   | ------------------ | --------- | ------------ | -------- | --------- | ------------- |
>   | halfcheetah-medium | 25.8±11.7 | **88.0±1.8** | 56.0±3.6 | 48.5±21.2 | 81.2±2.9      |
>   | walker2d-medium    | 34.8±10.6 | 1.5±0.7      | 3.3±2.1  | 2.5±3.5   | **53.5±14.8** |
>   | hopper-medium      | 1.2±0.8   | 65.3±6.3     | 0.0±0.0  | 0.0±0.0   | **74.2±5.7**  |
>
>   **The success rate of CDE and baselines with cut-off percentile=50**
>   |                    | BCQ      | CQL          | IQL      | TD3+BC    | CDE (Ours)   |
>   | ------------------ | -------- | ------------ | -------- | --------- | ------------ |
>   | halfcheetah-medium | 73.7±2.5 | **99.3±0.9** | 93.5±1.1 | 37.0±24.1 | 92.5±2.8     |
>   | walker2d-medium    | 38.0±1.2 | 47.6±17.1    | 18.7±2.5 | 15.7±10.8 | **67.0±5.0** |
>   | hopper-medium      | 9.3±1.9  | **90.0±0.8** | 6.0±7.8  | 0.0±0.0   | 88.0±5.7     |
>
>   When the percentile decreases, the reward becomes denser, and the threshold of success is lower. Therefore, the most baselines have higher success rate. We notice that CDE keeps relative high performance with 90 cutoff percentile. Meanwhile, CDE still obtains comparable performances with 50 percentile (CQL performs well in original 75 percentile setting).
>
> - **Q2. The performance on dense reward settings.**
>
>   We directly adopt the reported scores of baselines from original paper or D4RL benchmark and compare CDE with them.
>   |                           | BCQ       | CQL      | IQL      | TD3+BC   | CDE           |
>   | ------------------------- | --------- | -------- | -------- | -------- | ------------- |
>   | halfcheetah-medium        | 40.7      | **49.1** | 47.4     | 48.3     | 43.3±2.9      |
>   | walker2d-medium           | 54.5      | 82.9     | 78.3     | **83.7** | 73.8±4.4      |
>   | hopper-medium             | 53.1      | 64.6     | **66.3** | 59.3     | 51.2±3.7      |
>   | halfcheetah-medium-expert | 64.7      | 85.8     | 86.7     | **90.7** | 75.6±7.2      |
>   | walker2d-medium-expert    | **110.9** | 109.5    | 109.6    | 110.1    | 107.7±10.4    |
>   | hopper-medium-expert      | 57.5      | 102.0    | 91.5     | 98.0     | **108.6±4.8** |
>
>   While CDE may not outperform the baselines in dense reward settings, it achieves a significantly higher success rate when reward is sparse, which indicates that the baseline performances are predominantly reliant on dense rewards, thereby highlighting the effectiveness of CDE for sparse-reward offline data.
>
> References:
>
> [1] Provably good batch off-policy reinforcement learning without great exploration
>
> [2] Bellman-consistent pessimism for offline reinforcement learning
>
> [3] Conservative q-learning for offline reinforcement learning
>
> [4] Mildly conservative Q-learning for offline reinforcement learning

---

### Official Review · Reviewer_HRnW · 2023-11-04

**Soundness:** 3 good
**Presentation:** 3 good
**Contribution:** 3 good
**Rating:** 6
**Confidence:** 3

**Summary:**

This paper proposes a new offline RL method, called Conservative Density Estimation (CDE). It builds upon the DICE work by utilizing the state-action occupancy stationary distribution. However, the DICE type work requires the assumption on the concentrability, and the boundedness of the IS ratio for stable training. This paper tries to address this by introducing pessimism on the state-action stationary distribution, via a similar principle like conservative Q-learning.

**Strengths:**

- This paper points out the two type of approaches for offline RL, conservative Q-value based, which might perform worse under sparse reward setting due to iterative Bellman update; and DICE based, which implicitly relies on the concentrability. The proposed method nicely combines these two approaches, via a pessimism approach on the state-action stationary occupancy, which is a neat idea especially on the sparse reward setup.

- The paper is very well-written and easy to read. The method is very well-motivated, and clearly described, from policy evaluation to policy extraction. It also shows theoretical guarantees regarding to the concentrability bound, as well as performance gap.

- The experiments on sparse reward settings and small data regime is interesting, and support the CDE's effectiveness, especially at the sparse reward setup.

**Weaknesses:**

- Some parts of the method need to be dived into more details, for example, the CDE policy update happens when the value function converges. This also marks a difference compared with classical actor-critic methods, is it a key component in improving the empirical performance?

- More hyper-parameters introduced by the algorithm, such as the max OOD IS ratio, the mixture coefficient, etc. The ablation study on the mixture coefficient is interesting, demonstrating the robustness and giving some empirical guidelines on how to pick them. However, the study of max OOD IS ratio, though interesting, it does not provide any meaningful guidelines on how to choose them for various environments and applications.

- A clear understanding between CDE and CQL is missing, one is pessimism on the state-action state-action occupancy stationary distribution, the other is on the value function. Figure 2(d) only probably only shows the performance of CQL for a fixed pessimism-coefficient, it would be great to compare these two methods in detail, both theoretically and empirically.

**Questions:**

See Weakness.

---

> ### Author Response · Authors · 2023-11-16
> **Response to Reviewer HRnW**
>
> We thank the reviewer for valuable review and address the concerns as follows.
>
> - **W1. The effectiveness of "warm-up" stage for value function training.**
>
>   We use "warm-up stage" to refer to "policy update happens when the value function converges". We adopt warmup because (1) the value function is not well learned in the early stage and thus it cannot well guide the policy learning at that time, and (2) warm-up also reduces the computation overhead because policy update is stopped.
>
>   Meanwhile, the warm-up stage does not influence the final performances significantly. We give an ablation study on maze2d tasks (w. warm-up means stopping policy update at the beginning; w.o. warm-up means training value and policy together from start) and present results as follows:
>   | Task          | w. warmup  | w.o. warmup |
>   | ------------- | ---------- | ----------- |
>   | maze2d-umaze  | 134.1±10.4 | 136.2±13.2  |
>   | maze2d-medium | 146.1±13.1 | 141.7±17.3  |
>   | maze2d-large  | 210.0±13.5 | 205.4±15.4  |
>
>
> - **W2. More parameter studies. The study of max OOD IS ratio does not provide guideline.**
>
>   We include more parameter studies in appendix B.4 in original manuscript. We can observe that the performance of CDE is not sensitive to the most hyperparameters except max OOD IS ratio and mixture coefficient. For mixture coefficient, we can simply choose $\zeta$ to be close to 1 (e.g., set $\zeta=0.9$) by analysis in fig.3.
>
>   For the max OOD IS ratio, we agree that this study does not directly offer guidance on selecting $\tilde{\epsilon}$ when encountering new tasks. Nonetheless, by consistently employing the same $\tilde{\epsilon}$ across all experiments, we demonstrate its broad applicability to a variety of tasks. Meanwhile, our parameter study in appendix B.4.1 also shows that CDE is relatively resilient to the change of $\tilde{\epsilon}$. But we also admit that we may need to fine-tune it for specific new tasks.
>
>
> - **W3. Compare CDE and CQL in detail, both theoretically and empirically.**
>
>   Thanks for your suggestions.
>
>   As the reviewer mentioned, CDE and CQL apply pessimism on state-action occupancy distribution and Q-value function respectively. But in theory, we can also link them by the eq.(7), which shows that the optimal IS ratio is related to the value function. The main difference is that CQL adds an expectation of Q value on OOD region as a soft regularization with coefficient $\alpha$ while CDE enforces a direct constraint on IS ratio (can be also viewed as a constraint on advantage value by eq.(7)), which allows for more precise control on conservativeness for CDE.
>
>   We also provides some empirical results on maze2d-large task*. We test the performances of CQL with different regularization coefficient $\alpha$.
>   | $\alpha$     | 0.1      | 0.5  | 1.0      | 5.0     | 10.0    |
>   | ------------ | -------- | ---- | -------- | ------- | ------- |
>   | maze2d-large | -0.1±1.1 | 98.2±24.8 | 57.8±8.1 | 4.5±3.7 | 0.4±3.0 |
>
>   We can observe that CQL does achieve high score with $\alpha=0.5$. However, compared to CDE (see table 10 in appendix for parameter study of CDE), CQL's performance exhibits extreme sensitivity to the choice of $\alpha$. This sensitivity creates a challenging balance in fine-tuning the hyperparameter, oscillating between over-pessimism and substantial extrapolation error in OOD regions.
>
>   (*) We directly adopt results of CQL ($\alpha=5.0$) from D4RL benchmark in our original manuscript, which is a little different from the re-run results with $\alpha=5.0$.
>
> Thanks again and we hope our response addresses your main concerns.

---

### Official Review · Reviewer_voqr · 2023-11-10

**Soundness:** 3 good
**Presentation:** 3 good
**Contribution:** 3 good
**Rating:** 8
**Confidence:** 3

**Summary:**

In this work the authors propose CDE, an offline algorithm that builds upon a similar formulation as that of the DICE method of Nachum et al. In DICE the state-action distribution is optimized in order to maximize the reward under that distribution subject to an f-divergence regularization between the visit distribution and the empirical distribution under observed data. In additiona this optimization problem is constrained such that the state-action visit distribution is a valid discounted distribution under the initial-state and transition model of the MRP. Where CDE departs from DICE is in introducing an additional constraint such that the state-action distribution function evaluated at any state-action is less than a multiplicative factor of epsilon of some alternative distribution mu defined to have support only over out-of-distribution actions, and where mu is restricted to have the same marginal state distribution as the empirical distribution. Given this formulation the state-action distribution can then be optimized and the policy implied by this distribution can be extracted from the distribution. Additionally the authors extend the f-divergence to include an f-divergence with an extended distribution which mixes the empirical and mu distribution which a trade-off factor.

Based on these results the authors present some theoretical results bounding the importance ratio and bound the performance gap. The authors also provide a number of results on maze2d tasks and sparse mujoco tasks, predominantly showing that their method outperforms other comparable offline RL algorithms.

**Strengths:**

Overall the paper was quite good and the algorithm was well presented, albeit quite dense. The theoretical results do add to the paper (although I'm not sure how much) and the results were quite good and well presented.

**Weaknesses:**

Overall the paper was good and from my perspective merits inclusion at ICLR. My only real criticism of the work lies in its presentation, which although good I felt was dense. And the precise differences with e.g. DICE could have been made more clear, i.e. I would have liked to see more discussion of the effects of the additional constraint. To that end I'm not sure how much was added by the theoretical contributions of this work which could have been rather spent on explaining these differences at a higher level. And to be clear, I think the theorem(s) are useful, but since the proof itself and some of the underlying assumptions could not be included due to space constraints I'm not sure if these would not have been better moved into the appendix as a whole.

**Questions:**

See above.

---

> ### Author Response · Authors · 2023-11-16
> **Response to Reviewer voqr**
>
> We thank the reviewer for valuable review and address the main concerns as follows.
> - **Precise differences with DICE could have been made more clear; more discussion of the effects of the additional constraint.**
>
>   We summarize the key differences of CDE from previous DICE method:
>
>   1. In theory, CDE has an upper bound for concentrability ratio on OOD regions (Proposition 3). We also derive the upper bound in context of function approximation (Theorem 1). Therefore, the additional constraint avoids the arbitrarily large IS ratio and can mitigate issues caused by extrapolation error on OOD regions.
>   2. We further derive the performance gap bound in Theorem 2, which is absent in previous DICE paper.
>   3. Empirically, we verify the effectiveness of additional constraint on various tasks. The OOD constraint can reduce potential extrapolation error so we study the situations where offline data are scarce (i.e., the occupied state-action pairs are sparse in the whole space) and thus the OOD issue is more severe. From fig.1, we can find that CDE significantly outperforms optidice when offline data are limited.
>
> - **The theorem part can be better organized, e.g., to make the statement of theorem, assumption, and proof as a whole.**
>
>   Thanks for your suggestion. As you mentioned, we may not be able to insert proof into main text. So we repeat the statement of theorems in appendix to make them more readable and we also keep them in main test. The modification can be found in appendix part in revision.
>
> Thanks again for your feedback and we will emphasize them in the paper. We are more than glad to clarify it if you have further questions.
>
> References:
>
> [1] Nachum, Ofir, and Bo Dai. "Reinforcement learning via fenchel-rockafellar duality." arXiv preprint arXiv:2001.01866 (2020).
>
> [2] Lee, Jongmin, et al. "Optidice: Offline policy optimization via stationary distribution correction estimation." International Conference on Machine Learning. PMLR, 2021.

---

### Author Response · Authors · 2023-11-20
**General response**

Dear reviewers,

We are pleased to acknowledge that our paper has been recognized for its solid motivation (HRnW, MKs2) and novel method (6pFf), sound theoretical arguments (voqr, HRnW, 6pFf), impressive experiment results in sparse-reward tasks (voqr, HRnW, 6pFf), and clear presentation (voqr, HRnW). We also deeply appreciate the valuable feedback and guidance provided, which have been crucial in enhancing our work. Accordingly, we have made the following modifications in our revised manuscript:

- The sentences in the introduction have been rephrased for greater precision. (MKs2)
- Statements of propositions and theorems are now included in the appendix to enhance readability for the proofs. (voqr)
- An explanation of Slater's condition related to the original constrained optimization problem has been added to Appendix A.1. (6pFf)
- Additional experimental results on Mujoco and Adroit tasks have been presented in Appendix B.3.2 (MKs2); we have also incorporated an ablation study on the warm-up stage in Appendix B.4.4 (HRnW).

We hope these revisions can address your concerns. As we approach the end of the rebuttal phase, we want to inquire if you have any other questions regarding our response. We are willing to engage in the further discussion.

Sincerely,

Authors.

---

### Meta-Review · Area_Chair_aYtK · 2023-12-06

**Metareview:**

This paper addresses the assumption of boundedness of the IS ratio in DICE method and introduces a pessimism approach on the state-action stationary occupancy. It combines the pessimistic Q-learning and DICE approaches with a solid theoretical support. Experiments show good performance on the sparse reward setting.

Reviewers appreciate its solid motivation, theoretical contribution, and the advantage over commonly used baselines including BCQ, CQL and DICE variants.

Most concerns from the initial reviews of voqr, HRnW and 6pFf have been addressed with additional explanation and experiments in the rebuttal.

The main concern from reviewer MKs2 is on its empirical performance, lack of test tasks and baselines. "A more complex method must show significantly better results to justify its complexity". The authors provide an additional range of experiment results according to the reviewer' request. Reviewer MKs2 agrees the proposed CDE outperforms baselines in the Maze2D and sparse-MuJoCo domains, but does not show significant gap on the more challenging Adroit domain compared to value-based approaches, particularly compared to MISA and OneStep RL (cref Table 2 of [1]). The reviewer is concerned on CDE's advantage on more complex tasks.

Nonetheless, based on the current experiment results, CDE does show competitive performance in all of the three domains. It would be very useful to the authors to include discussion on lack of experiments in more complex sparse-reward environments such as Antmaze and Franka Kitchen due to its limitation that requires same initial state distribution in training and testing. Also, although CDE is designed for sparse offline RL setting, it is also important to state the fact that CDE may not outperform the baselines in dense reward settings as shown in the authors' response to MKs2.

The authors also discuss the remaining concerns/shortcoming on the choice of the hyperparameter of max IS ratio and how to satisfy the Slater's condition. Those should not be held against the acceptance of the submission.

**Justification For Why Not Higher Score:**

Incremental improvement on a well-know algorithm. Remaining concerns on the diminishing advantage of the prosed method in more complex domains.

**Justification For Why Not Lower Score:**

Good motivation, theoretical justification and sufficient experiment results.

---

### Decision · Program_Chairs · 2024-01-16

Accept (poster)